# NeuroPath: Neurobiology-Inspired Path Tracking and Reflection for Semantically Coherent Retrieval

**Junchen Li**[1], **Rongzheng Wang**[1], **Yihong Huang**[1], **Qizhi Chen**[1],
**Jiasheng Zhang**[1], **Shuang Liang**[1]*

[1]Institute of Intelligent Computing,
University of Electronic Science and Technology of China, Chengdu, China
junchenli@std.uestc.edu.cn, shuangliang@uestc.edu.cn

## Abstract

Retrieval-augmented generation (RAG) greatly enhances large language models (LLMs) performance in knowledge-intensive tasks. However, naive RAG methods struggle with multi-hop question answering due to their limited capacity to capture complex dependencies across documents. Recent studies employ graph-based RAG to capture document connections. However, these approaches often result in a loss of semantic coherence and introduce irrelevant noise during node matching and subgraph construction. To address these limitations, we propose NeuroPath, an LLM-driven semantic path tracking RAG framework inspired by the path navigational planning of place cells in neurobiology. It consists of two steps: *Dynamic Path Tracking* and *Post-retrieval Completion*. Dynamic Path Tracking performs goal-directed semantic path tracking and pruning over the constructed knowledge graph (KG), improving noise reduction and semantic coherence. Post-retrieval Completion further reinforces these benefits by conducting second-stage retrieval using intermediate reasoning and the original query to refine the query goal and complete missing information in the reasoning path. NeuroPath surpasses current state-of-the-art baselines on three multi-hop QA datasets, achieving average improvements of 16.3% on recall@2 and 13.5% on recall@5 over advanced graph-based RAG methods. Moreover, compared to existing iter-based RAG methods, NeuroPath achieves higher accuracy and reduces token consumption by 22.8%. Finally, we demonstrate the robustness of NeuroPath across four smaller LLMs (Llama3.1, GLM4, Mistral0.3, and Gemma3), and further validate its scalability across tasks of varying complexity. Code is available at https://github.com/KennyCaty/NeuroPath.

## 1 Introduction

Humans and other animals possess fundamental spatial navigation and episodic memory capabilities that enable accurate target searching. These abilities originate from cognitive maps formed through the coordinated activity of multiple neural systems within the hippocampus and entorhinal cortex. Place cells, as the cornerstone of spatial memory [29], work in concert with grid cells and boundary vector cells to support not only physical spatial navigation but also the construction of abstract memory spaces, organizing episodic memory [28, 8]. These neurons exhibit a continuous and interconnected activation pattern, transitioning from single-location encoding to path integration and broader cognitive map formation, dynamically supporting goal-directed behaviors [1].

---

*Corresponding Author

This work is supported by National Natural Science Foundation of China No.62406057, the Fundamental Research Funds for the Central Universities No.ZYGX2025XJ042, and the Sichuan Science and Technology Program under Grant No.2024ZDZX0011.

In knowledge-intensive tasks, current large language models (LLMs) resemble a navigator without a cognitive map. Despite their strong textual embedding capabilities for local contextual understanding [2], they struggle to form global associations within open-domain knowledge. Naive retrieval-augmented generation (RAG) adopts a strategy of individually embedding document chunks and retrieving relevant content based on similarity [21]. It is essentially a linear search for discrete knowledge. This strategy cannot capture the associations between knowledge, causing information silos when faced with complex queries that require multi-hop reasoning. Although iter-based RAG expands the knowledge coverage through a generate-retrieve-regenerate iteration, it lacks explicit modeling of the associations between knowledge and is still difficult to handle complex queries. Recently, researchers have proposed a new paradigm for knowledge organization and retrieval called graph-based RAG, which captures explicit associations across documents. However, existing graph-based methods have limitations: (1) the loss of semantic coherence clues and (2) irrelevant noise in node matching or subgraph construction. HippoRAG [14] leverages a knowledge graph (KG) and the Personalized PageRank (PPR) algorithm to propagate node importance. However, it does not explicitly use relationship semantics, leading to retrieval results that tend to structural relevance rather than path semantic coherence, as shown in Figure 1-(a). LightRAG [13] adopts dual-level retrieval to match nodes and construct subgraphs, enhancing integrated information retrieval capabilities. However, collecting direct neighbors in the subgraph construction brings considerable noise, as shown in Figure 1-(b).

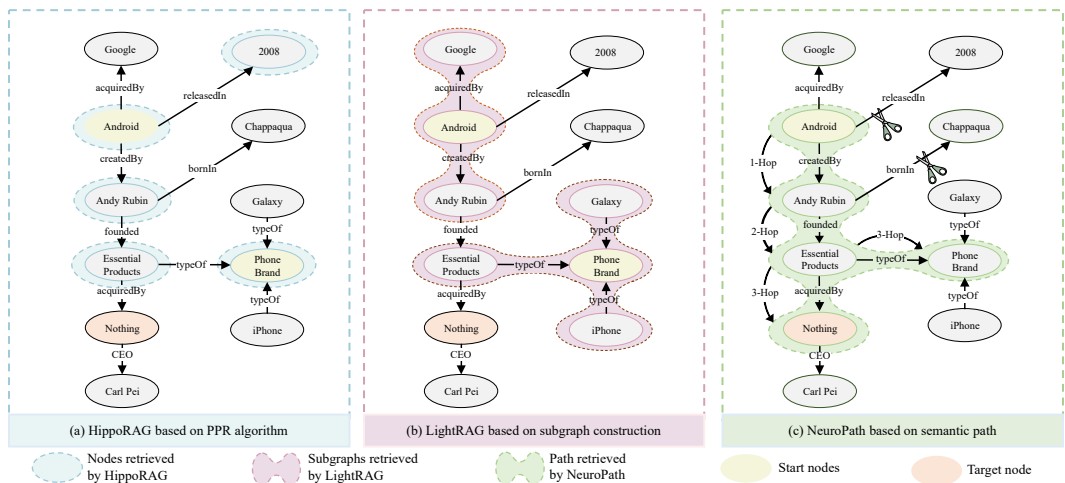

Figure 1: Comparison between graph-based and path-based RAG methods. To the query: *Which company acquired the phone brand created by the Android founder?* (a) HippoRAG uses the PPR algorithm to propagate node importance but ignores edge semantics, increasing the risk of retrieving incorrect nodes such as *2008*; (b) LightRAG's subgraph construction tends to introduce considerable noise; (c) Our method leverages coherent semantic paths for goal-directed tracking, progressively eliminating noise and tracking the correct answer *Nothing*.

We argue that an important advantage of modeling with graph structure is its explicit semantic reasoning paths formed through entity association. However, current graph-based methods focus more on structural relevance rather than semantic coherence, which weakens the semantic alignment with the query and introduces irrelevant noise. Existing methods rely on static topological modeling of knowledge associations, fundamentally different from the goal-directed dynamic encoding mechanisms of the human brain in complex environments or episodic memory. As shown in Figure 2, studies have shown that the brain's hippocampal place cells activate in specific regions as animals explore their environment [29], like a neural map. During navigation, the hippocampus preplays these cell sequences to support goal-directed path planning. These sequences can be dynamically reorganized based on task goals and can even predict unknown paths [30]. In addition, hippocampus replays them during rest to consolidate memory.

Inspired by these neurobiological mechanisms, we propose NeuroPath, a RAG framework based on semantic path tracking. Unlike methods that simply aggregate discrete knowledge nodes, our approach dynamically constructs goal-directed semantic paths, as illustrated in Figure 1-(c). Based on the KG

extracted from source documents, NeuroPath models each entity as a place cell in semantic space and each knowledge triple as a place field in spatial navigation. Guided by the goal of the query, NeuroPath simulates the hippocampal preplay and replay mechanisms: during preplay, it dynamically constructs semantic paths from seed nodes and filters out noisy information; during replay, it revisits prior reasoning chain to support path information completion aligned with the query goal. Specifically, we propose **Dynamic Path Tracking**, a strategy that leverages LLM to automatically filter and expand semantic paths. This strategy continuously marks valid paths, selectively expands paths, and predicts potential path directions for pruning. To further improve retrieval quality, we also propose a **Post-retrieval Completion** strategy that performs a second-stage retrieval using intermediate

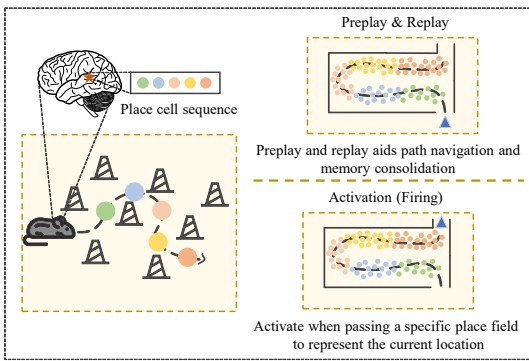

Figure 2: Place cells mechanism. Place cells represent specific spatial locations. During navigation, they preplay upcoming sequences, and during rest, they replay these to support memory consolidation.

reasoning generated by the LLM to complete missing information along the reasoning path.

In summary, the contributions of this work are as follows: (1) Inspired by neurobiology mechanisms, we novelly map the hippocampal place cell navigation and memory consolidation mechanisms into RAG, enhancing path reasoning and cross-document knowledge integration. (2) We propose NeuroPath, a RAG framework that dynamically tracks semantic paths to align with the query goal by leveraging LLMs and integrates intermediate reasoning to fill in missing information, thereby enriching semantic coherence and reducing noise. (3) NeuroPath achieves a new state-of-the-art performance on multi-hop QA, demonstrates robustness across LLM scales and tasks of varying complexity, and outperforms closed-source models using only a fine-tuned 8B open-source model.

## 2 Related Work

**From single-step to multi-step RAG**. The RAG framework greatly enhances LLMs' performance in knowledge-intensive tasks [10, 9]. Naive RAG methods retrieve top-k documents based on vector similarity between queries and document embeddings [21], but they overlook associations between documents. Due to the flat structure of vector databases, these methods cannot support multi-hop reasoning. As queries grow more complex and relevant knowledge becomes finer-grained, their effectiveness declines. Some methods refine queries to better capture complex intent and improve similarity matching [25, 6], yet they still rely on flat knowledge organization and struggle with multi-hop reasoning. Iterative retrieval RAG methods (iter-based RAG) interleave retrieval and generation to optimize queries for the next retrieval step [31, 35, 39], but they do not explicitly model relationships among knowledge entities, making it hard to connect information across documents.

**From flat to structured retrieval**. Recent works have advanced the ability of LLMs to reason over graph structures [32, 23, 41, 40]. To better capture document dependencies, researchers have proposed graph-based RAG methods [7, 13, 14, 34]. Unlike flat structures, these methods represent text as nodes and edges, enabling dependency propagation and multi-hop reasoning. Some methods model global and abstract information. For example, GraphRAG [7] uses community detection to summarize each group, while LightRAG [13] extracts local and global keywords to retrieve relevant nodes and edges. However, these methods focus on sense-making tasks, requiring a global understanding of knowledge and providing abstract summaries or diverse answers, which can introduce significant noise during the retrieval process. Some other methods focus on factual multi-hop QA, such as HippoRAG [14] and KG-Retriever [5]. However, these works pay more attention to structural relevance and collect documents by retrieving important nodes without explicitly utilizing edge semantics. Although HippoRAG 2 [15] improves query matching by replacing node matching with triple matching, it still suffers from the random walk nature of the PPR algorithm during the retrieval phase, which can lead the search to focus on incorrect nodes. These methods ignore path semantic coherence, resulting in retrieval errors or noise.

**From subgraph to path**. To avoid redundancy from subgraph construction, recent work extracts key paths directly through path-based methods. PathRAG [3] retrieves query-related nodes, constructs

paths using a flow-based pruning algorithm for resource allocation and scoring, and selects reliable paths. However, it allocates resources equally across outgoing edges, ignoring edge importance and semantics, which can misalign scores with actual meaning. As PathRAG focuses on sense-making and uses a different approach, we consider our method based on dynamic semantic path construction fundamentally distinct and better suited for multi-hop QA.

# 3 Methodology

## 3.1 Overview

The workflow of our framework consists of three specific steps, as shown in Figure 3, namely **Static Indexing**, **Dynamic Path Tracking**, and **Post-retrieval Completion**. We use an LLM to extract entities and relations from each document at once to build a KG index. Each entity builds a coreference set, and each triple segment is used as the smallest unit of the path receptive field. When retrieving, we simulate the preplay and replay mechanisms of place cells to perform path navigation and memory consolidation. Specifically, the LLM starts from seed nodes and selects valid paths from candidates based on their relevance to the query, while deciding whether to expand additional paths to track deeper information. If expansion is needed, it generates expansion requirements to guide the direction and prune irrelevant paths. Once the final paths are determined, we collect the source documents along the paths as context for question answering. This process is similar to place cells generating a pre-activation sequence during navigation and tracking the target through this sequence. We extract the reasoning generated by the LLM during path tracking and perform a second-stage retrieval by combining this generation and the original query. This simulates the replay mechanism in place cells, where sequences are reactivated during rest to consolidate memory.

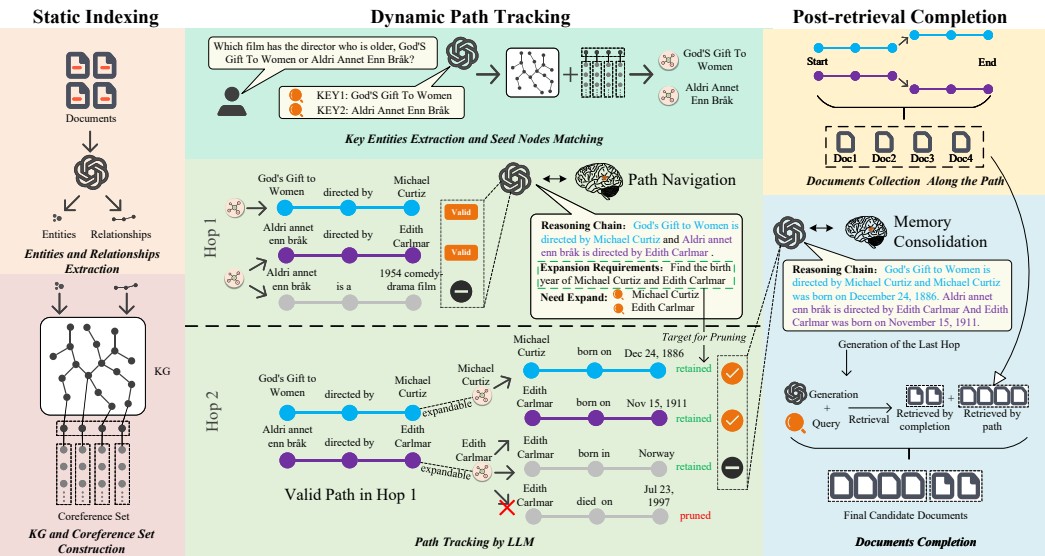

Figure 3: The overview of NeuroPath's workflow: (1) **Static Indexing**. Use an LLM to extract entities and relationships to build KG, and build a coreference set for each entity. (2) **Dynamic Path Tracking**. Using an LLM for goal-directed path tracking. The expansion requirements will be used for pruning. (3) **Post-retrieval Completion**. Collect documents along the path and leverage intermediate reasoning for second-stage retrieval to complete missing information in the reasoning path.

## 3.2 Static Indexing

Given a set $\mathcal{D} = \{d_1, d_2, \ldots, d_N\}$ containing $N$ documents, we use an LLM as a generator $G$ to extract the KG, aiming to capture information from the documents as detailed as possible. Specifically, we let $G$ extract the entity set $\mathcal{E}$ and the relation triple set $\mathcal{T}$ for each document $d_i$ at once.

In order to reduce the risk of missing references between semantically similar entities in the knowledge graph, we introduce a pseudo coreference resolution mechanism based on similarity. For each entity $e_i$ in KG, we construct its potential coreference set $R_i$ by including similar candidate entities whose vector cosine similarity exceeds the threshold of 0.8. We use an embedding model to uniformly encode entities and calculate similarity. The specific similarity calculation formula and coreference association set are as follows:

$$\text{Sim}(i, j) = \text{CosSim}(\text{Enc}(i), \text{Enc}(j)) \tag{1}$$

$$\mathcal{R}_i = \text{argtopk}_j \text{Sim}(i, j), \quad i, j \in \mathcal{E}, \tag{2}$$

where $\text{Enc}(\cdot)$ is an embedding model, and the argtopk operation retrieves the top-k entities based on similarity. By default, we retrieve five most similar entities as a coreference set.

### 3.3 Dynamic Path Tracking

**Seed nodes filtering rules**. Given a query $q$, we use an LLM as a generator $G$ to extract a set of key entities $\mathcal{Q} = G_{ent}(q)$, and select a set of the most similar nodes by $\text{Sim}(\cdot, \cdot)$ function from the indexing graph as initial seed nodes $\mathcal{S}$. Then expand the entities of the corresponding coreference association set $\mathcal{R}$ to become a new set $\mathcal{S}^0 = \{s + \mathcal{R}_s \mid s \in \mathcal{S}\}$, to better cover the path starting nodes.

In the subsequent path expansion, we directly use the expandable nodes of the paths and follow the above steps to add coreference nodes as seeds.

$$\mathcal{S}^h = \{s + \mathcal{R}_s \mid s \in \text{Link}(\mathcal{P}_{exp}^h)\}, \tag{3}$$

where $\text{Link}(\cdot)$ is a function to return the expandable nodes that connect to the path. $h$ is the current expanded hop and $\mathcal{P}_{exp}$ is the set of paths that need expansion.

**Path expansion rules**. In the specific expansion step, we retrieve the triple segments $\mathcal{P}_{sub}^h \subseteq \mathcal{T}$ connected to the seed nodes $\mathcal{S}^h$ and concatenate the current path set under expansion $\mathcal{P}_{exp}^h$. We then combine the current valid paths $\mathcal{P}_{val}^h$ to form the candidate paths for the next pruning and tracking.

$$\mathcal{P}_{cur}^{h+1} = \mathcal{P}_{val}^h + \text{Cat}(\mathcal{P}_{exp}^h, \mathcal{P}_{sub}^h), \quad \mathcal{P}_{exp}^0 = \mathcal{P}_{val}^0 = \emptyset, \tag{4}$$

where $\text{Cat}(\cdot, \cdot)$ denotes the operation of concatenating the triple segment to the end of the path under expansion. We merge the previous valid path set and the expanded path set into a new candidate path set. LLM can choose to keep or delete these paths to improve fault tolerance.

**Tracking by LLM**. We design a prompt to guide the LLM for path tracking, as shown in Appendix I. At each step of the tracking, we send the current candidate paths to LLM for filtering. LLM forms an intermediate reasoning chain according to the current valid information and mark the valid paths, determine whether expansion is needed and generate specific expansion requirements.

In order to make path expansion more directional and avoid exponential growth, we use expansion requirements generated by the LLM in the previous hop to prune paths before tracking.

$$\mathcal{P}_{cur}^{h'} = \text{argtopk}_p \text{Sim}(g^{h-1}, p), \quad p \in \mathcal{P}_{cur}^h \tag{5}$$

$$c^h, \mathcal{P}_{val}^h, g^h, \mathcal{P}_{exp}^h, ct = G_{tracker}(\mathcal{P}_{cur}^{h'}), \tag{6}$$

where $c$ is the reasoning chain organized under the current valid paths, $g$ is the specific requirement or goal of the expansion, $\mathcal{P}_{val}^h$ is the set of valid paths marked by the current hop, and $\mathcal{P}_{exp}^h$ is the set of paths that need to be expanded at the current hop. $ct$ is the flag for whether to continue to expand. The default setting for pruning top-k is 30. If $ct$ is 0, it means that the answer is found or no additional information can be added, otherwise continue to expand.

### 3.4 Post-retrieval Completion

After determining the final paths, we collect source documents along the paths as candidate documents $\mathcal{D}_p$. In addition, we adopt an enhanced second-stage retrieval step. Specifically, we concatenate the intermediate generation from the last hop of tracking and the original query as a new query to perform a second-stage retrieval to complete the candidate documents, in order to refine the query

goal and avoid the missing of path information. We define the new $q'$ as $q + c_{last} + g_{last}$, where $c_{last}$ and $g_{last}$ represent the reasoning chain and expansion requirements generated by the LLM in the last expansion hop (If the paths doesn't find an answer or reaches the maximum expansion hop count, expansion requirements will still be generated). They are concatenated with the original query $q$ for second-stage retrieval based on text similarity to improve the completeness of the information. The final set of source documents $\mathcal{D}_{ret} = \mathcal{D}_p \cup \mathcal{D}_e$, where $D_{ret}$ represents the final candidate document set, and $D_e$ represents the document set for second-stage retrieval.

## 4 Experiments

We conducted extensive experiments to evaluate NeuroPath and answer the following research questions (RQ): **RQ1**: How effective is NeuroPath? **RQ2**: How does NeuroPath demonstrate its advantages in semantic coherence and noise reduction? **RQ3**: Do all parts of our framework work? **RQ4**: What is its scalability to other small open-source models and to tasks of varying complexity?

### 4.1 Setup

**Datasets**. We selected three challenging multi-hop question answering datasets to evaluate our method: MuSiQue [38], 2WikiMultiHopQA [16] and HotpotQA [43]. All three datasets are used to evaluate open-domain multi-hop reasoning tasks, ranging from 2-hop to longer-hop reasoning scenarios. We followed the settings of HippoRAG [14], selected 1,000 questions from each dataset for evaluation, used all the documents from each selected dataset as the retrieval corpus. For each question, only a small number of supporting documents are involved to verify the retrieval performance. Additionally, to evaluate the model's robustness on tasks of varying complexity for RQ4, we utilize two simple QA datasets (PopQA [24], Natural Questions [19]) and a more challenging long-document multi-hop QA dataset (MultiHop-RAG [36]).

**Baselines**. We construct comparisons from (1) naive RAG methods BM25 [33], BGE-m3 [4], and Contriever [18]; (2) iter-based methods IRCoT [39], Iter-RetGen [35]; (3) graph-based methods HippoRAG [14], HippoRAG 2 [15], LightRAG [13]; (4) path-based method PathRAG [3].

**Metrics**. We report retrieval performance on recall@2 and recall@5 (R@2 and R@5 below), and question answering performance on exact match (EM) and F1 score.

**Implementation Details**. In our experiments, all graph indexing is performed using GPT-4o-mini [17]. To evaluate the performance and robustness of using large language models for path tracking, we further replace the retrieval component with a variety of open-source models. In the main experiment, we use GPT-4o-mini and Qwen-2.5-14B [42], and additional experiments with smaller-scale models are presented in Section 4.5. For NeuroPath, we set the maximum number of reasoning hops to 2 and use a Zero-Shot prompting setup. For IRCoT and Iter-RetGen, we set the maximum number of iterations to 3 and follow the original paper settings, with the number of documents retrieved per iteration set to 2, 4, 6 and 5, respectively. We evaluate all baselines using Contriever and BGE-M3 as retrievers. Additionally, since LightRAG and PathRAG do not directly return source documents, we do not assess its retrieval metrics. More details can be found in Appendix C.

### 4.2 Main Results (RQ1)

**Retrieval Results**. As shown in Table 1, NeuroPath outperforms all naive and graph-based baselines, including the state-of-the-art HippoRAG 2, with average improvements of 16.3% in Recall@2 and 13.5% in Recall@5. Compared to iter-based baselines, it also achieves average improvements of 8.6% in Recall@2 and 10.2% in Recall@5, demonstrating more stable and competitive performance. Additionally, our method reduces the average token consumption by 22.8% compared to iter-based baselines, indicating a more efficient use of resources. Detailed cost and efficiency comparison can be found in Appendix F. Our relatively lower performance on HotpotQA is primarily due to its lower knowledge integration requirements. This limits the advantages of the multi-hop reasoning methods. We further discuss this in Appendix E.3. In contrast, the MuSiQue dataset is more complex in its construction, designed for more difficult multi-hop reasoning, and our method achieves the best performance on it. Notably, our method maintains stable performance across different retrievers, while iter-based methods and HippoRAG 2 are highly sensitive to retriever choice, showing up to a

20% gap on 2WikiMultiHopQA. The above results demonstrate NeuroPath's superior performance in handling complex multi-hop queries. To further validate NeuroPath's applicability across different scenarios, we further evaluate its effectiveness on simple queries, see the Section 4.5. The results show that NeuroPath maintains competitive performance even on simpler tasks.

Table 1: Retrieval performance.

| Category | Method | MuSiQue | | 2Wiki | | HotpotQA | | Average | |
|---|---|---|---|---|---|---|---|---|---|
| | | R@2 | R@5 | R@2 | R@5 | R@2 | R@5 | R@2 | R@5 |
| Naive | BM25 | 32.2 | 41.2 | 51.7 | 61.9 | 55.4 | 72.2 | 46.4 | 58.4 |
| | Contriever | 34.8 | 46.6 | 46.6 | 57.5 | 57.1 | 75.5 | 46.2 | 59.9 |
| | BGE-M3 | 40.4 | 54.2 | 64.9 | 71.8 | 71.8 | 84.7 | 59.0 | 70.2 |
| **GPT-4o-mini (Contriever)** | | | | | | | | | |
| Iter-based | Iter-RetGen | 46.0 | 59.8 | 62.1 | 76.5 | **78.3** | **90.6** | 62.1 | 75.6 |
| | IRCoT | 42.2 | 55.8 | 54.2 | 70.0 | 68.6 | 83.3 | 55.0 | 69.7 |
| Graph-based | HippoRAG | 41.6 | 54.2 | 71.6 | 89.6 | 61.0 | 78.5 | 58.1 | 74.1 |
| | HippoRAG 2 | 41.8 | 55.5 | 62.5 | 74.2 | 65.3 | 83.4 | 56.5 | 71.0 |
| Path-based | **NP(Zero-Shot)** | **48.0** | **62.7** | **77.2** | **92.5** | 75.6 | 90.4 | **66.9** | **81.9** |
| **Qwen-2.5-14B (Contriever)** | | | | | | | | | |
| Iter-based | Iter-RetGen | 45.8 | 58.8 | 62.2 | 75.6 | **78.2** | **90.2** | 62.1 | 74.9 |
| | IRCoT | 40.2 | 52.7 | 56.2 | 72.0 | 65.9 | 76.9 | 54.1 | 67.2 |
| Graph-based | HippoRAG | 41.3 | 53.8 | 67.1 | 85.6 | 59.3 | 76.7 | 55.9 | 72.0 |
| | HippoRAG 2 | 40.3 | 52.7 | 56.7 | 67.7 | 63.6 | 81.3 | 53.5 | 67.2 |
| Path-based | **NP(Zero-Shot)** | **51.4** | **65.3** | **76.6** | **92.1** | 76.0 | **90.9** | **68.0** | **82.8** |
| **GPT-4o-mini (BGE-M3)** | | | | | | | | | |
| Iter-based | Iter-RetGen | 46.3 | 60.6 | 74.1 | 87.1 | 81.0 | 90.4 | 67.1 | 79.3 |
| | IRCoT | **48.7** | 62.7 | **79.0** | 92.0 | **81.3** | 90.8 | **69.7** | 81.8 |
| Graph-based | HippoRAG | 41.5 | 52.3 | 70.6 | 87.7 | 62.0 | 77.7 | 58.0 | 72.6 |
| | HippoRAG 2 | 43.6 | 61.2 | 72.2 | 88.8 | 71.1 | 89.4 | 62.3 | 79.8 |
| Path-based | **NP(Zero-Shot)** | 47.7 | **64.7** | 78.1 | **95.1** | 78.0 | **92.7** | 67.9 | **84.1** |
| **Qwen-2.5-14B (BGE-M3)** | | | | | | | | | |
| Iter-based | Iter-RetGen | 49.9 | 64.8 | 78.3 | 93.4 | **85.4** | **93.6** | **71.2** | 83.9 |
| | IRCoT | 45.0 | 56.4 | 76.6 | 91.8 | 78.5 | 84.7 | 66.7 | 77.6 |
| Graph-based | HippoRAG | 41.2 | 51.5 | 65.6 | 82.5 | 60.0 | 75.5 | 55.6 | 69.8 |
| | HippoRAG 2 | 44.7 | 59.9 | 70.5 | 85.1 | 72.8 | 89.1 | 62.6 | 78.0 |
| Path-based | **NP(Zero-Shot)** | **50.6** | **67.4** | **78.3** | **94.6** | 77.9 | 91.9 | 68.9 | **84.6** |

**QA Results**. As shown in Table 2, We report the QA performance using GPT-4o-mini and Contriever, results with BGE are similar and can be found in Appendix C. Our method outperforms all baselines on the MuSiQue and 2WikiMultiHopQA datasets, and performs slightly below HippoRAG 2 on HotpotQA. This is because Hotpot allows shortcuts answers through guessing [38]. We discuss the reasons for this observation in Appendix E.3. In addition, we found that LightRAG and PathRAG performed poorly on QA tasks. Despite leveraging graph structures to capture knowledge connections, they failed on multi-hop factual QA, performing worse than even naive methods. Appendix D details their limitations and explains why our path-based method is more effective than PathRAG.

Table 2: QA performance with Contriever as the Retriever.

| Category | Method | MuSiQue | | 2Wiki | | HotpotQA | | Average | |
|---|---|---|---|---|---|---|---|---|---|
| | | EM | F1 | EM | F1 | EM | F1 | EM | F1 |
| Naive | BM25 | 20.7 | 30.7 | 44.2 | 48.1 | 43.3 | 55.6 | 36.1 | 44.8 |
| | Contriever | 22.3 | 32.1 | 38.5 | 43.7 | 44.4 | 57.5 | 35.1 | 44.4 |
| | BGE-M3 | 27.8 | 39.7 | 48.2 | 54.5 | 47.8 | 61.8 | 41.3 | 52.0 |
| Iter-based | Iter-RetGen | 29.9 | 43.6 | 51.5 | 62.2 | 48.7 | 62.2 | 43.4 | 56.0 |
| | IRCoT | 30.0 | 42.1 | 47.7 | 57.0 | 45.2 | 59.9 | 41.0 | 53.0 |
| Graph-based | LightRAG | 4.2 | 13.7 | 10.8 | 19.4 | 14.7 | 27.7 | 9.9 | 20.3 |
| | HippoRAG | 27.8 | 40.5 | 58.6 | 69.1 | 43.3 | 57.8 | 43.2 | 55.8 |
| | HippoRAG 2 | 27.4 | 39.1 | 46.0 | 55.3 | **50.7** | **65.5** | 41.4 | 53.3 |
| Path-based | PathRAG | 8.4 | 20.8 | 21.0 | 31.6 | 23.8 | 41.2 | 17.7 | 31.2 |
| | NeuroPath | **31.4** | **44.3** | **63.4** | **73.2** | 50.5 | 64.7 | **48.4** | **60.7** |

## 4.3 Case Studies (RQ2)

To illustrate how NeuroPath addresses semantic incoherence and noise in graph-based methods, we compare it with several representative baselines, including HippoRAG 2 and LightRAG. As shown in Figure 4, we conduct case studies using a question from the MuSiQue dataset to evaluate the retrieval processes and answers of each method.

NeuroPath answers the question correctly by selectively expanding paths that preserve semantic coherence at each hop, effectively aligning with the question and avoiding irrelevant content. The documents retrieved rank the supporting evidence at the top, highlighting the quality of its retrieval process. In contrast, HippoRAG 2 applies PPR over both document and entity nodes but overemphasizes irrelevant content due to ignoring edge semantics. LightRAG retrieves a large subgraph (60 entities, 169 relations) but still fails to answer correctly, as much of the information is irrelevant to the question.

---

Query: Who was in charge of the place where Castricum is located?        Gold answer: Johan Remkes

- - - - - - - - - - - - - - - - - - - - - - - - - - - - - - - - - - - - - - - - - - - - - - - - - - - - - - - - - - -

Method: NeuroPath
Keywords: [**Castricum**]
Path: **castricum** ➜ is a town in ➜ **north holland**; **north holland** ➜ has king's commissioner ➜ **johan remkes**
Retrieved documents: [**Castricum**, **North Holland**, Lordship of Frisia, De Baarsjes, Laurens Reael]
Answer: **Johan Remkes**

- - - - - - - - - - - - - - - - - - - - - - - - - - - - - - - - - - - - - - - - - - - - - - - - - - - - - - - - - - -

Method: HippoRAG 2
Keywords: [**Castricum**]
Retrieved documents (Top ranked documents): [**Castricum**, Theo van den Boogaard, The Beach (film), Glass Beach, Roman Republic]
Answer: King's Commissioner of North Holland

- - - - - - - - - - - - - - - - - - - - - - - - - - - - - - - - - - - - - - - - - - - - - - - - - - - - - - - - - - -

Method: LightRAG
High level keywords: [Leadership, **Castricum**, Geographical location],
Low level keywords: [Historical figures, Governance, Local authority, Municipality]
Query uses 60 entities, 169 relations
Answer: Castricum is governed as part of modern municipal structures

---

Figure 4: Case studies. Comparison between NeuroPath and the graph-based baselines.

## 4.4 Ablation Studies (RQ3)

In this section, we ablate various components of NeuroPath, analyzing the impact of pruning, Post-retrieval Completion, hop counts, and prompts on retrieval performance.

Table 3: Dissecting NeuroPath. $p$ denotes the number of paths retained after pruning. *expansion_req* and *current_chain* correspond to the expansion requirements and current chain in the prompt.

| | MuSiQue | | | 2Wiki | | | HotpotQA | | |
|---|---|---|---|---|---|---|---|---|---|
| | R@2 | R@5 | Token (k) | R@2 | R@5 | Token (k) | R@2 | R@5 | Token (k) |
| w/ Post-retrieval Completion | | | | | | | | | |
| w/o pruning | **48.7** | **63.8** | 2891 | 76.8 | 92.0 | 1883 | 75.7 | **90.8** | 2504 |
| default (p=30) | 48.0 | 62.7 | ↓ 45.7% | **77.2** | **92.5** | ↓ 8.4% | **75.9** | 90.1 | ↓ 39.8% |
| p=20 | 47.3 | 62.2 | ↓ 52.6% | 76.5 | 90.8 | ↓ 17.3% | 74.9 | 89.4 | ↓ 47.0% |
| w/o Post-retrieval Completion | | | | | | | | | |
| Contriever(baseline) | 34.8 | 46.6 | - | 46.6 | 57.5 | - | 57.1 | 75.5 | - |
| max hop=1 | 35.5 | 35.5 | - | 61.0 | 62.3 | - | 61.3 | 66.4 | - |
| max hop=2 | 41.8 | 45.3 | - | 73.6 | 84.1 | - | 67.5 | 71.4 | - |
| max hop=3 | 42.1 | 45.4 | - | 73.9 | 84.5 | - | 67.7 | 72.2 | - |
| w/o expansion_req* | 40.9 | 44.2 | - | 69.9 | 75.9 | - | 65.8 | 70.1 | - |
| w/o current_chain* | 39.9 | 44.2 | - | 71.5 | 82.9 | - | 65.7 | 71.0 | - |

As seen in Table 3 lines 4-6, we conduct a comparison of performance and token consumption before and after pruning, demonstrating the effectiveness of the proposed pruning method. With 30 paths

retained after pruning, the retrieval performance is nearly equivalent and even higher in some metrics, while token consumption decreases by 45.7%, 8.4%, and 39.8%, respectively. With 20 paths retained after pruning, the performance reduction is not obvious, but token consumption is reduced by almost half on the MuSiQue and HotpotQA datasets. At the same time, this result also indirectly verifies that there is a lot of noise in the graph structure that is irrelevant to answering queries, and our method can greatly reduce this noise.

As seen in Table 3 lines 9-11, we report the performance of path tracking using only the LLM after removing the Post-retrieval Completion strategy. Removing this strategy leads to a significant drop in recall@5, indicating that this strategy helps compensate for missing path information and improves retrieval performance. The retrieval performance improves progressively as the maximum expansion hop parameter increases from 1 to 2 to 3, with a particularly notable gain observed when increasing from 1 to 2. Given that most of the questions in the three datasets are 2-hop questions, the improvement from 2 to 3 is relatively small. It is important to note that the hop parameter does not directly correspond to the number of reasoning steps in multi-hop QA. In our method, even with a hop value of 1, multi-hop reasoning across documents can be achieved through the linking of head and tail entities in triples. Notably, even with a maximum hop count of 1, our method outperforms the naive RAG baseline using the same retriever on the recall@2 metric. This demonstrates that the direction-aware path filtering enables higher-quality retrieval under fewer document conditions.

As seen in Table 3 lines 12-13, we conduct experiments by removing the prompts that instruct the LLM to generate the current reasoning chain and the expansion requirements (i.e., using only the query for pruning instead of expansion requirements), respectively. The results show performance drops, indicating two key insights: (1) Explicitly prompting the LLM to generate reasoning chains enhances its ability to assess path filtering and expansion; (2) Pruning with expansion requirements better captures semantic information related to path expansion, enabling more accurate direction calibration and noise suppression compared to query-based pruning.

## 4.5 Robustness to Model Scales and Task Complexity (RQ4)

In this section, we verify NeuroPath's robustness to different scales of LLMs and task complexity.

For LLMs, we evaluate its performance on various smaller LLMs, including Llama-3.1-8B-Instruct [12], GLM-4-9B-0414 [11], Mistral-7B-Instruct-v0.3 [27], and Gemma-3-4b-it [37]. We also fine-tune Llama-3.1-8B-Instruct [12] to evaluate potential performance gains. All experiments use Contriever as the retriever and adopt a One-Shot setting for NeuroPath.

Table 4: Retrieval performance of alternative small open-source LLMs on MuSiQue.

| | Llama-3.1-8B | | GLM-4-9B | | Mistral-7B | | Gemma-3-4B | | Average | |
|---|---|---|---|---|---|---|---|---|---|---|
| Method | R@2 | R@5 | R@2 | R@5 | R@2 | R@5 | R@2 | R@5 | R@2 | R@5 |
| Iter-RetGen | 44.0 | **58.4** | 45.1 | 58.9 | **41.8** | **54.1** | 42.1 | 55.3 | 43.3 | 56.7 |
| IRCoT | 40.3 | 53.7 | 40.2 | 49.1 | 38.5 | 50.9 | 36.8 | 42.4 | 39.0 | 49.0 |
| HippoRAG | 39.7 | 52.0 | 41.6 | 53.8 | 38.4 | 50.8 | 36.0 | 46.6 | 38.9 | 50.8 |
| HippoRAG 2 | 40.5 | 54.9 | 41.9 | 55.2 | 40.3 | 53.7 | 39.5 | 54.7 | 40.5 | 54.6 |
| **NeuroPath** | **45.5** | **58.4** | **46.6** | **59.3** | 40.1 | 53.9 | **43.4** | **58.1** | **43.9** | **57.4** |

We compare all the methods on the most challenging MuSiQue dataset, as shown in Table 4. NeuroPath consistently outperforms all baselines across the LLMs, except for Mistral-7B. Table 5 shows the performance improvement after fine-tuning on Llama-3.1-8B-Instruct. The results demonstrate that our method achieves a significant improvement through fine-tuning, even outperforming GPT-4o-mini. See the Appendix H for fine-tuning details.

Table 5: Retrieval performance after fine-tuning on LLama-3.1-8B-Instruct.

| | MuSiQue | | | 2Wiki | | | HotpotQA | | |
|---|---|---|---|---|---|---|---|---|---|
| Model | R@2 | R@5 | R@10 | R@2 | R@5 | R@10 | R@2 | R@5 | R@10 |
| GPT-4o-mini (default) | 48.0 | 62.7 | 70.8 | **77.2** | **92.5** | 94.1 | 75.6 | 90.4 | 94.3 |
| Llama-3.1-8B (SFT) | **50.5** | **64.9** | **72.1** | **77.2** | **92.5** | **94.2** | **76.4** | **90.8** | **94.5** |

To verify the adaptability of our method to simple queries (standard tasks), we conduct additional experiments on two simple QA datasets: PopQA [24] and NaturalQuestions [19], which mainly

consist of single-entity-centered questions. Following the setup of the main experiments, we randomly select 1000 queries from each dataset.

Table 6: Retrieval performance on standard tasks with Contriever and GPT-4o-mini.

| Method | PopQA R@2 | PopQA R@5 | NQ R@2 | NQ R@5 | Average R@2 | Average R@5 |
|---|---|---|---|---|---|---|
| Contriever | 27.0 | 43.2 | 29.1 | 54.6 | 28.1 | 48.9 |
| Iter-RetGen | 35.4 | 46.6 | **40.9** | **68.6** | 38.1 | **57.6** |
| IRCoT | 30.2 | 38.2 | 35.1 | 55.8 | 32.6 | 47.0 |
| HippoRAG | 36.5 | **52.7** | 22.5 | 45.7 | 29.5 | 49.2 |
| HippoRAG 2 | 34.2 | 47.4 | 32.1 | 59.5 | 33.2 | 53.4 |
| **NeuroPath** | **40.5** | 49.3 | 35.8 | 65.2 | **38.2** | 57.2 |

As shown in Table 6, our retrieval performance is still better than the graph-based method, and even better than the iter-based method on the PopQA dataset. The PopQA dataset is particularly entity-centric, with questions constructed around specific entities. Graph structure enhanced methods, which represent both entities and their relations, can quickly locate relevant entities for retrieval, giving them a natural advantage on this type of data.

Our main experiments rely on short, clean paragraphs, which may not reflect real-world scenarios involving long, noisy documents. To address this and further test NeuroPath's robustness, we evaluate it on the MultiHop-RAG [36] dataset. We sample 1000 questions for evaluation. Given that documents in this dataset are substantially longer (each document has become almost 20 times longer, see Appendix C), we chunk them into 512 token segments. This setup inherently challenges retrieval performance with fragmented evidence and noise, providing a more stringent evaluation. We compare NeuroPath against baselines from prior experiments.

Table 7: Retrieval performance on MultiHop-RAG Dataset with GPT-4o-mini.

| Method | MultiHop-RAG (Contriever) R@2 / R@5 | MultiHop-RAG (BGE) R@2 / R@5 |
|---|---|---|
| Contriever | 9.5 / 20.4 | - / - |
| BGE-M3 | - / - | 25.4 / 44.7 |
| Iter-RetGen | 17.8 / 30.9 | **28.8** / 45.8 |
| IRCoT | 8.8 / 17.1 | 25.3 / 39.8 |
| HippoRAG | 16.5 / 26.6 | 18.2 / 30.2 |
| HippoRAG 2 | 15.7 / 29.5 | 21.4 / 40.8 |
| **NeuroPath** | **23.7 / 39.0** | 26.1 / **46.8** |

As shown in Table 7, our method outperforms typical baselines, especially when using the smaller Contriever. Notably, other methods still exhibit sensitivity to the choice of embedding model, whereas NeuroPath shows relatively stable performance differences between Contriever and BGE-M3.

## 5    Conclusions

In this paper, we introduced NeuroPath, a novel RAG framework inspired by hippocampal place cell mechanisms to address the critical limitations of semantic incoherence and noise in existing graph-based methods. Our core contributions, Dynamic Path Tracking and Post-retrieval Completion, empower LLMs to dynamically construct and refine reasoning paths that are semantically aligned with the query. Extensive experiments demonstrate that NeuroPath achieves new state-of-the-art performance on challenging multi-hop QA datasets. Furthermore, our findings confirm its robustness across various LLMs and task complexities. We underscore a key insight: for multi-hop reasoning, semantic coherence is a more effective retrieval principle than simple structural relevance. This work not only provides a high-performance solution for multi-hop QA but also opens a promising direction for developing more faithful, explainable, and neurobiologically inspired reasoning systems within the RAG paradigm.

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

# A    Neurobiology-inspired Mechanism

In this section, we will briefly introduce the mechanism of hippocampal place cells and cognitive maps as a supplement to explain our motivation.

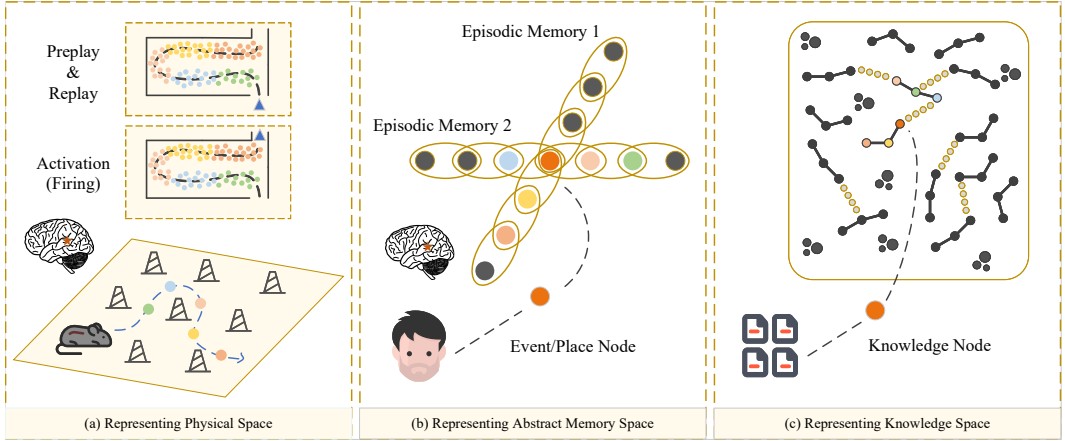

Figure 5: Representation forms of cognitive maps in physical and memory space, and their analogy to knowledge representation.

**Place cells in the hippocampus**. Place cells are found in the CA1 and CA3 regions of the hippocampus and show place-specific firing when animals explore their environment [29]. They play an important role in spatial positioning and navigation, as shown in Figure 5-(a). When an animal reaches a specific location during movement, a corresponding place cell becomes active. The area represented by each place cell is called a place field. Groups of place cells work together with grid cells and boundary vector cells in the entorhinal cortex to form a cognitive map and support spatial processing [28]. Studies have shown that place cell activity sequences can replay previously experienced paths during rest or sleep. This phenomenon is believed to be related to memory consolidation and episodic memory [8]. Pfeiffer et al. confirmed that the preplay of place cell sequences plays an important role in predicting future paths [30]. During navigation, the hippocampus generates a sequence of place cell activity in chronological order, like a neural map. Even when the combination of start and target positions is completely new. Place cell sequences show flexibility and adaptability, allowing dynamic path planning based on the current environment and goals.

**Place cells and cognitive map**. Eichenbaum et al. believe that the hippocampus should participate in the representation of a wide range of memory spaces through place cells and collaborative cells, and the representation of physical space is only a special case of memory space representation [8]. This theory believes that cognitive maps are essentially episodic memories composed of sequences of different events or location nodes, and then different episodic memories are linked together through repeated or common elements (also called nodes) to establish a memory space. As shown in Figure 5-(b).

**Graph-structured knowledge organization**. Knowledge in the real world does not exist in isolation, but is interconnected through complex associative relationships. This organization of knowledge resembles the structure of cognitive maps, which can be represented as a network composed of nodes and edges, as illustrated in Figure 5-(c). Both path navigation in neurobiology and knowledge retrieval in multi-hop QA task, the core mechanism involves the integration of multi-source information and progressively locate the target. Therefore, inspired by the role of place cell sequences in navigational planning of future paths [30], we simulate the preplay and replay mechanisms of place cells, thereby enabling goal-directed knowledge retrieval and memory consolidation.

# B NeuroPath Workflow

Algorithm 1 outlines the NeuroPath workflow, which comprises three main stages: **Static Indexing**, **Dynamic Path Tracking**, and **Post-retrieval Completion**. Moreover, Figure 10 and Figure 11 provide illustrative examples of the Dynamic Path Tracking and Post-retrieval Completion processes, respectively.

---

**Algorithm 1** NeuroPath Framework Workflow

---

**Require:** Source documents $\mathcal{D}$ and query $q$
**Ensure:** Candidate documents $\mathcal{D}_{ret}$ (Final retrieved)

 1: **Static Indexing:**
 2: Extract entities and relation triples from documents $\mathcal{D}$ using LLM          See Fig. 12 for prompt
 3: Construct knowledge graph (KG) from extracted facts
 4: Construct coreference sets based on embedding similarity of entities
 5: **Dynamic Path Tracking:**
 6: Use LLM to extract key entities from query $q$          See Fig. 13 for prompt
 7: Retrieve initial seed nodes via entity similarity
 8: Expand seed nodes with coreference set
 9: **repeat**
10:    **if** hop $h = 0$ **then**
11:       Gather initial seed nodes connected triple segments from KG
12:       Concatenate these triple segments to form candidate paths
13:       Prune candidate paths using similarity with query $q$
14:    **else**
15:       Identify expandable nodes linked to current path and expand them with coreference set as new seed nodes
16:       Gather seed nodes connected triple segments from KG
17:       Concatenate these triple segments to form candidate paths
18:       Prune candidate paths using similarity with previous expansion goal $g^{h-1}$
19:    **end if**
20:    Retain top-30 candidate paths
21:    Use LLM to:          See Fig. 14 for prompt
22:       Select valid paths
23:       Mark paths that need to be expanded
24:       Generate reasoning chain based on valid paths $c^h$
25:       Generate next-hop expansion requirements $g^h$
26:       Decide whether to continue, represented by the generated flag $ct$
27: **until** $ct = 0$ or maximum hop reached
28: **Post-retrieval Completion:**
29: Collect all source documents from selected valid paths as $\mathcal{D}_p$
30: Concatenate the $c$ and $g$ from the final hop with the original query $q$ to form the new query $q'$
31: Retrieve additional documents $\mathcal{D}_e$ using $q'$
32: **return** final set $\mathcal{D}_{ret} = \mathcal{D}_p \cup \mathcal{D}_e$

---

# C Detailed Experimental Settings and Results

All open-source LLMs in our experiments are deployed using vLLM [20] on NVIDIA GeForce RTX 4090. For GPT-4o-mini, we use the official OpenAI API.

We show the detailed data of entity and relationship extraction when NeuroPath performs static indexing in Table 9. For MultiHop-RAG, we selected 1,000 questions from the original dataset and merged all supporting documents for each question into a single corpus. We then chunked all documents within this corpus into blocks of 512 tokens and retained the original title field. Table 8 shows the comparison of the average number of tokens in documents across different datasets.

Experiments for HippoRAG & HippoRAG 2, LightRAG, and PathRAG are conducted using the official code provided by the authors. LightRAG uses version 1.3.4 with the specified mode set to 'local', and all other baselines use default settings. Except for LightRAG and PathRAG, we

Table 8: Comparison of the average number of tokens per document.

| | MuSiQue | 2Wiki | HotpotQA | PopQA | NQ | MultiHop-RAG |
|---|---|---|---|---|---|---|
| Avg tokens / doc | 110 | 105 | 128 | 129 | 136 | 2,289 (chunk to 512) |

standardize the QA prompts across all models. Additionally, as LightRAG and PathRAG do not support retrieval performance evaluation, we follow their original configuration by merging each dataset into a single document and chunking. Moreover, to ensure consistency in QA output style with other methods, we add standardized output prompts to LightRAG and PathRAG, enabling fairer comparisons in terms of EM and F1 metrics. The detailed prompt is provided in Appendix I.

Table 9: Statistics of KG data extracted by GPT-4o-mini.

| | MuSiQue | 2Wiki | HotpotQA | PopQA | NQ | MultiHop-RAG |
|---|---|---|---|---|---|---|
| Entities | 104,442 | 49,362 | 89,947 | 87,233 | 92,852 | 39,281 |
| Relations | 40,861 | 15,049 | 33,914 | 22,163 | 40,273 | 18,564 |
| Triples | 120,226 | 60,188 | 108,621 | 106,061 | 116,689 | 43,638 |

In addition to the QA performance with Contriever, we supplement the QA performance comparison using BGE-M3 as the retriever in Table 10. Similar to the results with Contriever, our method consistently outperforms the baselines.

Table 10: QA performance with GPT-4o-mini and BGE-M3.

| | | MuSiQue | | 2Wiki | | HotpotQA | | Average | |
|---|---|---|---|---|---|---|---|---|---|
| Category | Method | EM | F1 | EM | F1 | EM | F1 | EM | F1 |
| Naive | BM25 | 20.7 | 30.7 | 44.2 | 48.1 | 43.3 | 55.6 | 36.1 | 44.8 |
| | Contriever | 22.3 | 32.1 | 38.5 | 43.7 | 44.4 | 57.5 | 35.1 | 44.4 |
| | BGE-M3 | 27.8 | 39.7 | 48.2 | 54.5 | 47.8 | 61.8 | 41.3 | 52.0 |
| Iter-based | Iter-RetGen | 31.7 | 46.3 | 56.0 | 66.7 | 49.8 | 64.5 | 45.8 | 59.2 |
| | IRCoT | 30.8 | 44.6 | 61.5 | 72.1 | 50.3 | 64.4 | 47.5 | 60.4 |
| Graph-based | LightRAG | 5.2 | 15.8 | 16.8 | 30.3 | 18.0 | 32.7 | 13.3 | 26.3 |
| | HippoRAG | 25.8 | 36.7 | 57.4 | 66.7 | 43.7 | 57.2 | 42.3 | 53.5 |
| | HippoRAG 2 | 31.6 | 44.5 | 57.9 | 67.8 | **55.0** | **70.1** | 48.2 | 60.8 |
| Path-based | PathRAG | 7.1 | 19.8 | 18.5 | 28.9 | 21.7 | 38.5 | 15.8 | 29.1 |
| | NeuroPath | **33.0** | **46.4** | **63.7** | **74.1** | 50.3 | 65.2 | **49.0** | **61.9** |

# D Compare to PathRAG

In the main experiment, NeuroPath demonstrates markedly different performance compared to PathRAG, due to two key factors: differences in path construction methods and task objectives.

**Difference in path construction methods**:

- NeuroPath leverages LLMs to dynamically filter and expand paths starting from selected seed nodes. Our approach emphasizes semantic coherence during path expansion and alignment with the query objective. Throughout the process, the expansion direction is continuously adjusted based on the query to minimize noise. In essence, the entire path expansion chain is dedicated to answering the query, retaining only semantically relevant and mutually reinforcing information that contributes to the answer.

- In contrast, PathRAG first selects key nodes and then constructs paths among them. Aside from the key nodes being related to entities in the query, the path construction process does not ensure semantic relevance to the query. Moreover, its path pruning strategy distributes resources evenly based on node connectivity, which often fails to build effective reasoning paths.

**Difference in task objectives**:

- NeuroPath is designed for multi-hop reasoning tasks that require strictly factual answers. This typically demands a precise alignment between the reasoning path and the query requirements.
- In contrast, PathRAG—like LightRAG—is tailored for sense-making tasks, which often involve answering global or abstract questions. These tasks generally benefit from collecting comprehensive and diverse information to support broad understanding. While PathRAG reduces the information redundancy introduced during subgraph construction in LightRAG, this optimization only targets redundant paths between nodes. It does not fundamentally address the issue of introducing query-irrelevant noise.

# E  Error Analysis

## E.1  Retrieval Results Distribution

We present the distribution of retrieval results in Figure 6, which focuses on the top-10 retrieved documents for each question, where the gray portion represents those that were not successfully retrieved.

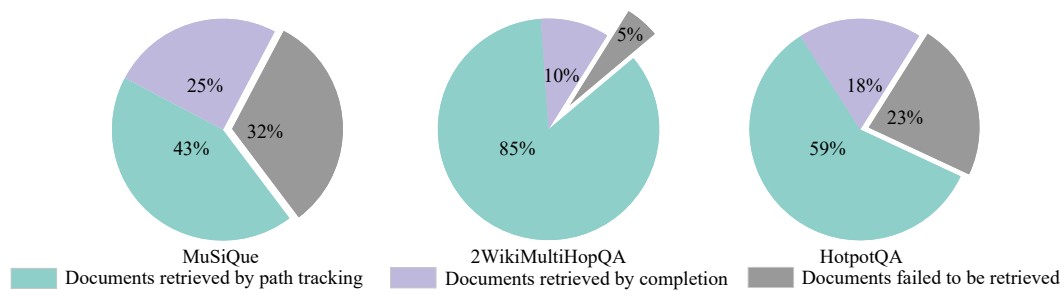

Figure 6: Retrieval results distribution.

## E.2  Nodes Mismatch

We analyze the questions in the MuSiQue dataset where the supporting documents were not perfectly retrieved. Among these, 31.5%, 40.8%, and 26.7% are categorized as 2-hop, 3-hop, and 4-hop questions, respectively. We attempted to increase the maximum length of path expansion, but found that it did not significantly improve performance. Therefore, we proceed to analyze aspects such as entity extraction and nodes matching.

Table 11: The degree of mismatch between the initial seed nodes and the nodes along the path with respect to the supporting document. 2Wiki* indicates randomly deleting one key entity from the query-extracted entity list (if the list has more than one entity).

| MuSiQue | Mismatch (%) | 2Wiki | Mismatch (%) | 2Wiki* | mismatch (%) |
|---|---|---|---|---|---|
| Seed nodes | 49.6 | Seed nodes | 36.1 | Seed nodes | 57.7 |
| Path nodes | 39.1 | Path nodes | 8.9 | Path nodes | 36.7 |

We analyzed node-to-supporting-document mismatch on the MuSiQue dataset (Table 11). A match is defined by the intersection between the node set and the set extracted from the supporting documents. Results show that 49.6% of initial seed nodes fail to match any supporting documents. While path expansion reduces the mismatch rate to some extent, many documents remain uncovered. For comparison, the initial seed nodes mismatch rate is 36.1% on the 2WikiMultiHopQA dataset, and path expansion reduces it further to 8.9%. This difference is likely due to the higher complexity of MuSiQue questions. Figure 7 shows an example of the extracted entities from MuSiQue. The model failed to identify the core event *Tripartite discussions*, leading to a mismatch in path tracking. Similarly, when we randomly remove one key entity from each query in 2WikiMultiHopQA, the

mismatch rate of initial seed and path nodes increases significantly, highlighting the importance of accurate initial key entities extraction.

> Query: What month did the Tripartite discussions begin between Britain, France, and the country where, despite being headquartered in the nation called the nobilities commonwealth, the top-ranking Warsaw Pact operatives originated?
>
> Extracted key entities: ['Britain', 'France', 'Warsaw Pact', 'nobilities commonwealth']

Figure 7: Example of query key entities extraction in MuSiQue dataset.

### E.3 Discussion of Underperformance on HotpotQA

In this section, we discuss the reasons why both the retrieval and generation processes did not achieve optimal performance on the HotpotQA dataset.

Previous studies [14, 38, 26] have shown that many HotpotQA questions can be answered correctly without complete multi-hop reasoning, often via a naive RAG approach. This is primarily because: (1) HotpotQA documents are short and information-dense; (2) most questions are simple synthetic two-hop queries requiring minimal reasoning; and (3) its distractors are generally weak, which simplifies the retrieval of target passages. While iter-based methods perform well on HotpotQA by leveraging these characteristics to match dense chunks and refine queries, they are less efficient, consuming 22.8% more tokens (see Appendix F). Furthermore, their sensitivity to embedding models leads to inconsistent performance.

We observed a notable discrepancy in our experiments: HippoRAG 2 underperformed NeuroPath on retrieval metrics yet surpassed it on QA metrics. We attribute this to HotpotQA's allowance for shortcuts answers derived through guessing [38]. As shown in Figure 8, HippoRAG 2 retrieved only one of the two required supporting documents but was still able to infer the answer by spotting the mention of "Gal Gadot". This is because HippoRAG 2 matches triples and passages using the entire query rather than intermediate nodes, a method that favors information-dense documents and thus facilitates such guessing. In contrast, Table 12 demonstrates that NeuroPath is less reliant on this guessing. It achieves a 9.3% higher proportion of correct retrievals among its correct answers compared to HippoRAG 2. This is a direct result of NeuroPath's design: it expands paths one hop at a time, with each expansion being influenced by the semantic coherence of the previous path. This process inherently prevents sudden jumps to random nodes, ensuring a more faithful reasoning process.

Table 12: Proportion of correct retrievals among correct answers.

|            | Correct retrievals | Correct answers | Proportion (%) |
|------------|--------------------|-----------------|----------------|
| HippoRAG 2 | 408                | 507             | 80.4           |
| NeuroPath  | 453                | 505             | 89.7           |

## F   Cost and Efficiency Comparison

**Token Cost**. We compared the token consumption of different methods, which is closely related to LLMs, at each stage (on GPT-4o-mini), as shown in Figure 9. Our method consistently achieves the lowest token usage across all stages. It reduces token usage by 31.1% on average during graph indexing stage (LLM-based graph indexing), and by 22.8% on average during retrieval stage (LLM-based iterative methods). In the QA stage, our method (along with other methods) reduces token usage by 89.2% compared to LightRAG, indicating that LightRAG introduces significant irrelevant noise through subgraph construction. Our method filters this noise during path tracking, improving both retrieval and QA performance.

**Time Cost**. Table 13 compares the graph indexing time of NeuroPath with other methods requiring graph indexing on 2WikiMultiHopQA. Since we perform NER and relation extraction for each document within a single LLM call, the time consumption is reduced by nearly half compared to

Query: Which post DC Extended Universe actress will also play a role in what is intended to be the fifth installment of the DC Extended Universe?
Gold answer: Gal Gadot

Justice League (film)
...It is intended to be the fifth installment in the DC Extended Universe (DCEU). The film is directed by Zack Snyder and written by Chris Terrio and Joss Whedon, from a story by Snyder and Terrio, and features an ensemble cast that includes Ben Affleck, Henry Cavill, Gal Gadot, Jason Momoa, Ezra Miller, Ray Fisher, Ciarán Hinds, Amy Adams, Willem Dafoe, Jesse Eisenberg, Jeremy Irons, Diane Lane, Connie Nielsen, and J. K. Simmons...

Aquaman (film)
...

List of DC Extended Universe cast members
...

Batman v Superman: Dawn of Justice
...

DC Extended Universe
...

======================= Supporting documents not retrieved=======================

Wonder Woman (2017 film)
... It is the fourth installment in the DC Extended Universe (DCEU). The film is directed by Patty Jenkins, with a screenplay by Allan Heinberg, from a story by Heinberg, Zack Snyder, and Jason Fuchs, and stars Gal Gadot, Chris Pine, Robin Wright, Danny Huston, David Thewlis, Connie Nielsen, and Elena Anaya...

Figure 8: Example where HippoRAG 2 answered successfully but retrieval failed.

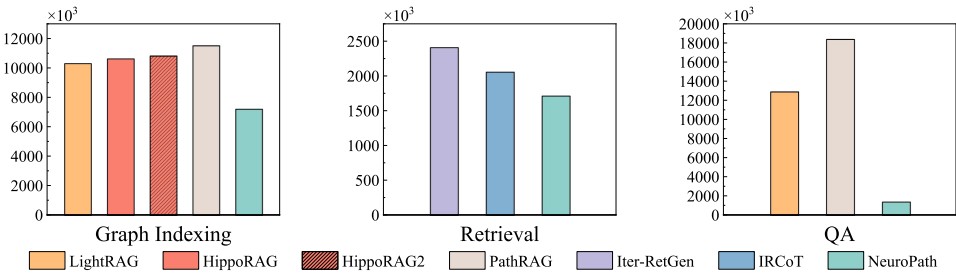

Figure 9: Comparison of Token Consumption across three stages. In QA, NeuroPath represents all methods other than LightRAG and PathRAG (all using the top 5 documents for answering).

other methods. The main bottleneck of the retrieval time cost of our method lies in the call waiting of LLM. The call time of our method on LLM is almost the same as that of other methods based on iterative retrieval using LLM. Table 14 presents a comparison of retrieval time consumption between NeuroPath and other iterative retrieval methods. Our slightly higher time cost mainly comes from an extra LLM call for extracting query's key entities. In addition, the pruning strategy requires the use of an embedding model to calculate the embedding of each path and the expansion requirement in parallel. It takes about 0.2s for each question on Contriever and BGE-M3, which is much shorter than the waiting time for LLM calls. Therefore, we believe that the time complexity of the pruning strategy is acceptable.

Table 13: Comparison of graph indexing time with graph-based methods.

|  | HippoRAG | HippoRAG 2 | LightRAG | PathRAG | NeuroPath |
|---|---|---|---|---|---|
| Time (minutes) | 64 | 62 | 52 | 72 | 39 |

Table 14: Comparison of average retrieval time per question with iter-based methods.

| | Iter-RetGen | IRCoT | NeuroPath |
|---|---|---|---|
| Time (seconds) | 5.6 | 4.7 | 6.7 |

## G  Limitations

Although our dynamic construction of semantic paths significantly improves performance on multi-hop QA tasks, several limitations remain. First, NeuroPath heavily relies on LLM calls for path filtering and expansion, which introduces a time bottleneck. Second, due to the low granularity of the triple segments expanded in each step, some simple questions may be unnecessarily split into multi-hop problems. Finally, as we do not optimize the retrieval corpus itself (e.g., via summarization), the method may face limitations in sense-making tasks that require abstraction or summarization.

## H  Fine-tuning Details

We additionally select 1,500 questions from the original 2WikiMultiHopQA dataset and follow the same procedure as in the main experiments using DeepSeek-V3 [22]. The outputs are then used to fine-tune Llama-3.1-8B-Instruct [12].

Specifically, we fine-tune Llama-3.1-8B-Instruct using QLoRA with 8-bit quantization and FlashAttention-2 for efficiency. LoRA is applied to all layers with a rank of 8, $\alpha = 16$, and dropout = 0. We train for 3 epochs using the AdamW optimizer and a cosine learning rate schedule with a base learning rate of 5e-5.

## I  LLM Prompts

Figure 12 and Figure 13 show the prompts when NeuroPath performs static indexing and extracts key entities from the query.

Figure 14 shows the prompt when NeuroPath performs path tracking, and Figure 15 shows the example of One-Shot.

Figure 16 shows the restrictive prompt words used by NeuroPath and other models for QA question answering to unify the model output style.

Query: Which film has the director born first, Sensation In Savoy or Devasuram?
Key entities: [Sensation In Savoy, Devasuram]

Hop1:

Paths:
0: sensation in savoy->was directed by->eduard von borsody;  <expandable>: [eduard von borsody]
1: sensation in savoy->was made by->bavaria film;  <expandable>: [bavaria film]
2: devasuram->directed by->i v sasi;  <expandable>: [i v sasi]
3: sensation in savoy->was made in the year->1950;  <expandable>: [1950]
4: devasuram->considered as->one of the best films of i v sasi s career;  <expandable>: [one of the best films of i v sasi s career]
5: sensation in savoy->is a film of genre->comedy crime;  <expandable>: [comedy crime]
6: sensation in savoy->starring->paul klinger;  <expandable>: [paul klinger]
…
Output:
{
    "current_chain": "The director of Sensation In Savoy is Eduard Von Borsody. The director of Devasuram is I V Sasi.",
    "valid_ids": [0, 2],
    "expansion_requirements": "Find when were Eduard Von Borsody and I V Sasi born.",
    "need_expand_ids": [0, 2],
    "continue": 1
}

Hop2:

Paths:
0: sensation in savoy->was directed by->eduard von borsody;
1: devasuram->directed by->i v sasi;
2: sensation in savoy->was directed by->eduard von borsody; eduard von borsody->born on->13 june 1898;
<expandable>: [13 june 1898]
3: devasuram->directed by->i v sasi; i v sasi->was born on->28 march 1948;  <expandable>: [28 march 1948]
4: sensation in savoy->was directed by->eduard von borsody; eduard von borsody->died on->1 january 1970;
<expandable>: [1 january 1970]
5: sensation in savoy->was directed by->eduard von borsody; eduard von borsody->was->austrian;
<expandable>: [austrian]
6: sensation in savoy->was directed by->eduard von borsody; eduard von borsody->was a->film editor;
<expandable>: [film editor]
...
Output:
{
    "current_chain": "The director of Sensation In Savoy is Eduard von Borsody, who was born on 13 June 1898. The director of Devasuram is I. V. Sasi, who was born on 28 March 1948.",
    "valid_ids": [2, 3],
    "expansion_requirements": "",
    "need_expand_ids": [],
    "continue": 0
}

Figure 10: Example of Dynamic Path Tracking.

Query: Which film has the director born first, Sensation In Savoy or Devasuram?

Valid Path:
1: sensation in savoy->was directed by->eduard von borsody; eduard von borsody->born on->13 june 1898;
2: devasuram->directed by->i v sasi; i v sasi->was born on->28 march 1948;

The source documents that the paths pass through (the following are the document titles):
Doc1: *Sensation in Savoy*; Doc2: *Eduard von Borsody*; Doc3: *Devasuram*; Doc4: *I. V. Sasi*

The chain of reasoning in the last hop:
The director of Sensation In Savoy is Eduard von Borsody, who was born on 13 June 1898. The director of Devasuram is I. V. Sasi, who was born on 28 March 1948.

Prepare a second retrieval to augment candidate documents and compensate for potential missing information in the paths:
New Query: Which film has the director born first, Sensation In Savoy or Devasuram? The director of Sensation In Savoy is Eduard von Borsody, who was born on 13 June 1898. The director of Devasuram is I. V. Sasi, who was born on 28 March 1948.

New documents retrieved (the following are the document titles):
Doc5: *Boaz Davidson*; Doc6: *Jacques Deray* ; ⋯

Supporting Document Titles:
Doc1: *Sensation in Savoy*; Doc2: *Eduard von Borsody*; Doc3: *Devasuram*; Doc4: *I. V. Sasi*

Figure 11: Example of Post-retrieval Completion.

Your task is to extract named entities from the given passage and construct an KG (Knowledge Graph) from the passage and the entities you extracted.
Requirements:
The entity type can be [organization, person, object, location, time, event, term], etc. And the KG should describe the information contained in the text as detailed as possible.
The format of a KG (Knowledge Graph) triple is ["head node", "relation", "tail node"], and each part must have a value.
Coreference and Pronoun Resolution:
Specific names should be explicitly resolved to maintain clarity.
Respond with a JSON Object.

# Example Begin
Passage:
```
Teutberga
Teutberga( died 11 November 875) was a queen of Lotharingia by marriage to Lothair II. She was a daughter of Bosonid Boso the Elder and sister of Hucbert, the lay- abbot of St. Maurice's Abbey.
```
Output:
```
{
    "named_entities": ["Teutberga", "11 November 875", "Lotharingia", "Lothair II", "Bosonid Boso the Elder", "Hucbert", "St. Maurice's Abbey"],
    "triples": [
        ["Teutberga", "died on", "11 November 875"],
        ["Teutberga", "was a queen of", "Lotharingia"],
        ["Teutberga", "married to", "Lothair II"],
        ["Teutberga", "is a daughter of", "Bosonid Boso the Elder"],
        ["Teutberga", "is a sister of", "Hucbert"],
        ["Hucbert", "is the lay-abot of", "St. Maurice's Abbey"]
    ],
}
```
# Example End

Passage: {Passage}
Output:

Figure 12: Prompt for entities and relationships extraction.

---

Please extract all named entities that are important for solving the questions below. Place the named entities in json format.

# Example 1
Question: Who wrote the book that was published earlier, "The Great Gatsby" or "To Kill a Mockingbird"?
Output :
{
    "named_entities": ["The Great Gatsby", "To Kill a Mockingbird"]
}
# Example 2
Question : Are the cities of Paris and Berlin located in the same country?
Output :
{
    "named_entities": ["Paris", "Berlin"]
}

Question: {Question}
Output:

Figure 13: Prompt for extracting key entities from a query.

To answer a given query, you need to select a set of valid clue paths to form a chain of reasoning and determine whether some paths need to be expanded with more information to help answer the query.

# Explanation
Valid paths can provide intermediate reasoning steps or evidence to help answer the query. Paths may be redundant, filter them out.
A path that contains the <expandable> tag is an expandable path, which identifies a phrase that can be expanded with additional information.
If information is sufficient or no more valid information can be added, stop expanding.
Return the required JSON object.

# JSON object format
{
    "current_chain": "A string. You should think step by step. Try to form the current chain of reasoning.",
    "valid_ids": [List of valid path IDs (int) (sort by helpfulness to query)],
    "expansion_requirements": "A string. If need expand, provide the specific requirements of the expansion. Otherwise set to an empty string.",
    "need_expand_ids": [List of path IDs (int) that need to be expanded. (if any)],
    "continue": 0 or 1 (0 = stop expanding, 1 = continue expanding)
}

Figure 14: Prompt for path tracking.

# Example
Query: Which film has the director who was born later, El Extrano Viaje or Love In Pawn?
Paths:
0: El Extrano Viaje->released in->1964;
1: El Extrano Viaje->directed by->Fernando Fernan Gomez; <expandable>: [Fernando Fernan Gomez]
2: El Extrano Viaje->starring->Jose Isbert; <expandable>: [Jose Isbert]
3: Love in Pawn->released in->1953;
4: Love In Pawn->directed by->Charles Saunders; <expandable>: [Charles Saunders]
Output:
{
    "current_chain": "The director of El Extrano Viaje is Fernando Fernan Gomez. And the director of Love In Pawn is Charles Saunders",
    "valid_ids": [1,4],
    "expansion_requirements": "Find when were Fernando Fernan Gomez and Charles Saunders born.",
    "need_expand_ids": [1,4],
    "continue": 1
}

Figure 15: One-Shot example for path tracking.

=========================== NeuroPath and other methods ============================

You are an assistant who is good at reasoning and question answering. You need to analyze the given information according to the questions. Response start after "Thought: ", where You will go through the reasoning process step by step to explain how you came to the conclusion. Conclude with "Answer: " to present a concise, definitive response, devoid of additional elaborations.

===================== Individual Settings of LightRAG and PathRAG =====================

(The following prompt needs to be appended to the QA prompt in LightRAG and PathRAG.)
At the end of your output, conclude with "Answer: " to present a concise, definitive response, devoid of additional elaborations.

Figure 16: Prompt for QA.

