# OpenReview forum: "NeuroPath: Neurobiology-Inspired Path Tracking and Reflection for Semantically Coherent Retrieval"
_NeurIPS.cc/2025/Conference — NeurIPS 2025 poster_

### Official Review · Reviewer_QcTf · 2025-06-20

**Clarity:** 1
**Significance:** 2
**Originality:** 3
**Rating:** 3
**Confidence:** 4

**Summary:**

The authors propose a method for improving RAG methods' ability to answer multi-hop questions involving dependencies across documents. The authors propose a neurobiology-inspired method. The authors validate the method on three multi-hop benchmarks MuSiQue, 2WikiMultiHopQA, and HotPotQA using 1k questions from each. The authors compare their proposed method against standard RAG, iter-based methods, and graph-based methods (HippoRAG).

**Questions:**

please see above.

**Ethical Concerns:**

["NO or VERY MINOR ethics concerns only"]

**Final Justification:**

I believe this manuscript would benefit from substantial revision to improve its accessibility, particularly for the general community at NeurIPS. In terms of validation, I find the method to invovled multiple complex components must be weighed against the performance gains (that are in some setting mixed). I'd encourage the authors to consider revisions to clarify the need for such complexity and improve the accessbility of the work.

**Limitations:**

I believe a more explicit discussion of the proposed method's limitations are warranted. As noted above, computational overhead, method complexity, and mixed results on HotpoQA should be discussed and presented in more detail.

**Quality:**

2

**Strengths And Weaknesses:**

* Motivation for what structural relevance is and why it's insufficient needs to be clarified in the introduction. I'd recommend including an example of the queries the users have in mind or some definitions with intuition to motivate the approach.
* Method claims to learn semantic paths that form a knowledge graph. It would be nice to provide additional analysis or evidence to support this claim, even if qualitative.
* Method involved several steps
    * 1) Static indexing: build set of entity, relations, documents. Use cosine similarity of embeddings to ensure quality and retain the five most similar entities., 2) Dynamic path tracking, and 3) Post-retrieval completion.
* I found exposition of the method and its many steps difficult to follow. In particular, the high level descriptions of steps were not easy to translate into algorithmic implementations for me.
* Similarly, as a researcher with little knowledge of biology and cognitive science, I found the motivation while admirable to be inaccessible. I imagine other readers without the necessary background would find it equally challenging and would benefit from a more accessible exposition.
* Iter-based methods seem to perform better on HotpotQA in many settings. Can the authors comment on why this may be the case? The proposed method need not outperform existing baselines across all benchmarks to be a worthwhile contribution, but an acknowledgement of the this limitation beyond "HotpotQA ... has lower knowledge integration requirements" and discussion of tradeoffs is missing in this current version.
* For question-answering, the authors call into the question the quality of HotpotQA noting it's prone to shortcut learning. If that's the case, I recommend the authors simply include HotpotQA in the appendix and note this limitation  and instead evaluate other benchmarks of quality.
* Given the mixed performance on HotpotQA, I also recommend teh authors expand the list of benchmarks to include additional multi-hop reasoning datasets. For example, the original IRCot paper includes several other suggestions https://arxiv.org/abs/2212.10509
* Missing baseline Multihop RAG https://arxiv.org/pdf/2401.15391 that seems particularly relevant to compare against (and is well cited).
* Nice to see the additional study using other model families.
* Compared to existing methods, how computationally intensive is the proposed method? It seems to involve several computations of similar and refinement that would be important to quantify any computational tradeoffs  compared to existing methods.

---

> ### Author Rebuttal · Authors · 2025-07-31
>
> Thank you for the detailed and insightful comments. We address the issues you raised below.
>
> ***Issue 1: Motivation clarification: structural relevance vs. semantic coherence.***
>
> We have already provided a detailed query example and textual explanation in Figure 1 (Page 2), using comparisons across different methods to illustrate our motivation. We understand that further clarification of certain concepts can help better convey the motivation.
>
> 1. We define **structural relevance** as methods that retrieve graph nodes based on the graph topology (e.g., Personalized PageRank in HippoRAG, subgraph construction in LightRAG). However, such approaches risk retrieving nodes that are structurally close but semantically irrelevant.
>
> 2. In contrast, **semantic coherence** requires that the path formed by nodes and edges aligns with the logical intent of the query(e.g.,[Android]->(createdBy)->[Andy Rubin]->(founded)->[Essential Products]->(acquiredBy)->[Nothing] as illustrated in Figure 1 (c)).
>
>
>
> ***Issue 2: Lack of analysis or evidence for semantic path learning***
>
> 1. The case study (Section 4.3, Pages 7–8) shows example path sequences selected by Neuropath. Compared to HippoRAG and LightRAG, NeuroPath contain richer relational information and show clear logic connections to the query.
>
> 2. Figure 10 and Figure 11 (Appendix B, Page 20) provide concrete examples of Dynamic Path Tracking and Post-retrieval Completion
>
> 3. The ablation study (Section 4.4, Page 8) compares retrieval performance using only path tracking (i.e., without post-retrieval completion). The table below is based on Table 3 (Page 8) in the main paper and highlights the relative improvement (%) from path tracking alone under the recall@2 metric.
>
> Table 1: Ablation study of the path tracking without post-retrieval completion. (Section 4.4, Page 8)
>
> || MuSiQue| 2Wiki  | HotpotQA |
> | - | - | - | - |
> | Contriever | 34.8| 46.6 | 57.1|
> | max hop=1  | 35.5 ($\uparrow$2.0%)  | 61.0 ($\uparrow$30.9%) | 61.3 ($\uparrow$7.4%)  |
> | max hop=2  | 41.8 ($\uparrow$20.1%) | 73.6 ($\uparrow$57.9%) | 67.5 ($\uparrow$18.2%) |
>
>
>
> ***Issue 3-4：The methods and motivation background are unclear and inaccessible.***
>
> * Methods：
>
> We describe our framework in detail in the following three aspects:
>
> 1. **Algorithm 1 (Appendix B, Page 14)** formalizes workflow with step-by-step pseudocode.
> 2. **Figure 3 (Page 4)** visually maps the interplay between static indexing, dynamic path tracking, and post-retrieval completion. Extended examples in **Figure 10-11 (Pages 20-21)** demonstrate concrete LLM outputs during path expansion. **Appendix J** (Page 19) provides detailed prompts.
> 3. **Mathematical specifications** (e.g., coreference resolution in Eq.1-2, path concatenation in Eq.4, pruning via Eq.5) anchor abstract concepts to computable operations.
>
> We argue this multi-angle presentation meets reproducibility standards for complex systems, and we have also released our code and detailed reproduction steps on an anonymous GitHub repository (linked in the abstract).
>
>
>
> * Motivation:
>
> To aid readers unfamiliar with neuroscience, we summarize the relevant biological mechanisms and their parallels to retrieval, with further details in Appendix A (Page 13).
>
> 1. Limitation of current graph-based RAG: Methods like HippoRAG (PageRank-based) and LightRAG (subgraph expansion) construct static topologies, often introducing semantically incoherent or noisy results. In contrast, brain retrieval is dynamic and goal-directed.
> 2. In the brain, *place cells* support two key mechanisms: **preplay** for path planning and **replay** for memory consolidation.
>     NeuroPath simulates this via LLM-based **dynamic path tracking** (navigation) and **post-retrieval completion** (consolidation).
> 3. Brain/Bio-inspired mechanisms have recently sparked new ideas in the LLM era. Our work aims to shift the RAG paradigm from **“brain-like storage”** (e.g., HippoRAG’s graph-based indexing) to **“brain-like reasoning”**, by enabling dynamic retrieval over structured knowledge graphs.
>
>
>
> ***Issue 5-6: Lack of trade-off discussion for the method’s underperformance on HotpotQA.***
>
> Thank you for pointing out this limitation. We discuss it from three perspectives: 1. Shortcuts in HotpotQA, 2. Iter-based methods and tradeoffs, and 3. NeuroPath’s tradeoffs.
>
> 1. Shortcuts in HotpotQA: Prior work (e.g., HippoRAG [1], MuSiQue [2]) has shown that HotpotQA often allows shallow shortcut-based answering without full multi-hop reasoning. We have analyzed it in Appendix F.3 (Page 17). HippoRAG 2 achieves 9.3% more correct answers not backed by full evidence compared to NeuroPath (Table 10, Page 17). In fact, many HotpotQA questions can be effectively retrieved and answered using naive RAG methods [3], due to several inherent limitations:
>    1. HotpotQA documents are short and dense.
>    2. Most questions are synthetically constructed 2-hop queries with limited dependency complexity [2].
>    3. Prior work [1] shows that distractor documents are often weak, making target passages easier to retrieve.
>
> 2. Iter-based methods and tradeoffs: These perform well on HotpotQA by matching dense chunks and refining queries via LLMs. However, they consume 22.8% more tokens (Appendix G) and are more sensitive to embedding models (Table 1, Page 7), resulting in inconsistent performance.
>
> 3. NeuroPath’s tradeoffs: Due to the limitations of HotpotQA, NeuroPath may underperform iter-based methods on this dataset. However, its path-based, fine-grained retrieval is better suited for complex multi-hop reasoning (e.g., MuSiQue and 2WikiMultiHopQA).
>
>    It is also more robust to embedding model changes due to shorter and cleaner path token. It is worth noting that we evaluate the scalability of NeuroPath on other tasks:
>
>    1. We demonstrate that our NeuroPath can retain the impressive performance of simple retrieval on the standard retrieval datasets PopQA and NaturalQuestions (Appendix D, Page 15).
>    2. We also extended our evaluation to the more complex and uncertain sense-making task. (follow the settings and baselines in HippoRAG 2 paper) Our test results on the NarrativeQA dataset are as follows, and NeuroPath still maintains good performance.
>
> Table 2: QA performance on sense-making task.
>
> |Method| NarrativeQA (F1 Score) |
> | - | - |
> | NV-Embed-v2 (7B) | 24.2 |
> | GraphRAG         | 20.9 |
> | LightRAG         | 9.0  |
> | HippoRAG         | 16.1 |
> | HippoRAG 2       | 25.2 |
> | **NeuroPath**    | **32.7**  |
>
> Given HotpotQA’s known limitations and NeuroPath’s broader scalability, we believe this trade-off is acceptable and will elaborate further in revised Appendix F.3.
>
>
>
> ***Issue 7-8：Suggest using higher-quality datasets such as MultiHop-RAG.***
>
> Thank you for the suggestion. We conducted additional experiments on this dataset.
>
> Setup: We selected 1,000 questions and split the documents into 512-token chunks. Contriever and BGE were used as embedding model, respectively.
>
> Table 3: Retrieval results on MultiHop-RAG.
>
> || **MultiHop-RAG (Contriever)** | **MultiHop-RAG (BGE)** |
> | - | - | - |
> |Method| R@2 / R@5 | R@2 / R@5 |
> | Contriever | 9.5 / 20.4  | -                      |
> | BGE-M3   | -  | 25.4 / 44.7  |
> | Iter-RetGen   | 17.8 / 30.9     | **28.8** / 45.8 |
> | HippoRAG 2    | 15.7 / 29.5                   | 21.4 / 40.8            |
> | **NeuroPath** | **23.7** / **39.0**           | 26.1 / **46.8** |
>
>  Results show that NeuroPath remains in the leading position. It is worth noting that other baselines show high sensitivity to the performance of the embedding model, whereas NeuroPath remains more robust in comparison.
>
>
>
> ***Issue 9：The computational cost needs clearer quantification and comparison to existing methods.***
>
> We rigorously evaluated the cost and efficiency of NeuroPath against baselines (Appendix G, Page 18-19), and our results demonstrate a favorable tradeoff between overhead and performance gains.
>
> Table 4: Comparison of graph indexing time with graph-based methods. (Page 19)
>
> |                | HippoRAG | HippoRAG 2 | LightRAG | PathRAG | NeuroPath |
> | - | - | - | - | - | - |
> | Time (minutes) | 64       | 62         | 52       | 72      | 39        |
>
> Table 5: Comparison of average retrieval time per question with iter-based methods. (Page 19)
>
> |                | Iter-RetGen | IRCoT | NeuroPath |
> | - | - | - | - |
> | Time (seconds) | 5.6         | 4.7   | 6.7       |
>
> 1. NeuroPath’s indexing is 1.5× faster than graph/path-based methods due to single-pass KG extraction. It reduces token usage by 31.1% on average (Appendix G, Figure 9).
>
> 2. NeuroPath’s retrieval is 1~2s slower per query than iter-based methods, this is mainly due to the additional query processing. Path pruning allows embeddings to be computed in parallel on the GPU, with an average runtime of just 0.2 seconds per query. By calling LLM twice for each query, our token cost is reduced by 22.8% (Appendix G, Figure 9).
>
>
> ***Limitation: Need more discussion of the proposed method's limitations and cost.***
>
> As we detailed in our response to Issues 5-9, we discussed the limitations of NeuroPath on HotpotQA, as well as the trade-offs between performance and scalability. We further validated the superiority of our method on a more challenging datasets MultiHop-RAG. Finally, we compared the computational cost of different methods and clarified the source of our method’s complexity.
>
> [1] HippoRAG: Neurobiologically Inspired Long-Term Memory for Large Language Models
>
> [2] MuSiQue: Multihop Questions via Single-hop Question Composition
>
> [3] Compositional Questions Do Not Necessitate Multi-hop Reasoning

---

> > ### Comment · Reviewer_QcTf · 2025-08-04
> >
> > Thank you for the comments and additional results.
> >
> > **The methods and motivation background are unclear and inaccessible.** while I appreciate "a multi-angle presentation" and definitions in the appendix, I do not find the presentation of the method to be sufficiently clear, accessible, or reproducible.
> >
> > **Computational tradeoffs** thank you for pointing out these comparisons. These address my concerns regarding the computation tradeoff. I would recommend briefly mentioning a reference to these results in the main paper.

---

> > > ### Author Response · Authors · 2025-08-05
> > > **Clarification of Motivation and Methodology**
> > >
> > > Thank you for your feedback. We would be glad to provide a brief clarification of our motivation and method once again.
> > >
> > > * Motivation: Our framework provides a solution to the recent research short-comings (incoherence and noise) by neurologically inspired path tracking and post-retrieval completion. Unlike HippoRAG/LightRAG/PathRAG, our method simulates dynamic, goal-directed reasoning in the brain to enhance semantic coherence and query alignment in retrieval, shifting from static brain-like storage to dynamic brain-like reasoning.
> > >
> > > * Method: We use a prompt-driven LLM for dynamic path tracking and prune the number of input paths to reduce token usage. Documents along the final path are collected as candidates. The last-hop output of the LLM (including the reasoning chain and expansion requirements) is used for a secondary retrieval to fill in missing content. (The necessary details are fully provided in Section 3 Page 4 (Methodology).)
> > >
> > > We have provided a detailed query example and explained the biological mechanisms we simulate and how they map to our method. (Figure 1 and Figure 2, Page 2 in the original Introduction section). **These directly address the concern raised in the original review: “recommend including an example of the queries … or some definitions … to motivate the approach”.**  Additionally, the detailed prompts (App. J) and open-source code ensure reproducibility of the method and results. For any questions on the workflow, we provide pseudocode and example inputs and outputs (App. B) for clarification.
> > >
> > > Other reviewers (e.g., Reviewer y7M1, hmsH, and GVGa) have explicitly noted that our method is “well-motivated” (GVGa), “easily reproducible“ and "with properly motivated and analyzed cases” (y7M1), and “easy to understand” (hmsH). **These comments suggest that the framework is accessible, clearly presented, and reproducible.**
> > >
> > > In response to your feedback (Issues 1–9), we have provided additional explanations and experimental results, and we hope these clarifications will address your concerns. We would be grateful if you could reconsider your evaluation in light of these clarifications. If any parts of the motivation or method remain unclear, please let us know which specific parts are confusing, and we would be happy to clarify further.

---

### Official Review · Reviewer_GVGa · 2025-07-03

**Clarity:** 4
**Significance:** 4
**Originality:** 3
**Rating:** 5
**Confidence:** 5

**Summary:**

In this work, the authors propose NeuroPath, a retrieval framework that draws inspiration from place cells and their path integration capabilities to improve the performance of current graph-based RAG systems such as HippoRAG, LightRAG and PathRAG on multi-hop retrieval benchmarks. NeuroPath uses a combination of Dynamic Path Tracking and Post-Retrieval Completion in order to accomplish more faithful traversal of an automatically extracted knowledge graph. Dynamic Path Tracking involves using an LLM to decide which paths to continuously expand into across the KG from an initial set of of entities. In contrast, Post-Retrieval Completion filters irrelevant paths, thus avoiding an explosion in the subgraph visited during search. This methodology accomplishes strong performance over other advanced graph-based RAG methods and iterative retrieval based methods on three multi-hop QA datasets.

**Questions:**

N/A

**Ethical Concerns:**

["NO or VERY MINOR ethics concerns only"]

**Final Justification:**

The authors have addressed all of my concerns. This paper should be accepted.

**Limitations:**

Yes

**Quality:**

3

**Strengths And Weaknesses:**

Strengths:
- Neurobiological inspiration is interesting and insightful
- Their methodology is well-motivated and the problem they are addressing is important and timely.
- The approach, although relatively simple, addresses a well known problem with graph-based RAG systems in a principled way.
- Their discussion is comprehensive, strong ablation studies and case studies demonstrate why and how their method performs well.

Weaknesses:
- The experimental setting is relatively sound, however, there are two somewhat serious issues:
1) In the abstract and introduction, the authors explain that their method "surpasses the state-of-the-art" but I do not find this statement convincing given that simple large-scale retrieval embeddings are known to perform better than the reported results on the three benchmark datasets (HippoRAG 2). Given that the authors use several 6-9B parameter LLMs for generation, it seems that they could have used large-scale embedding models which are simple and very powerful.
2) Although strong performance on multi-hop retrieval is important, it would make for a much stronger empirical analysis if other RAG benchmarks were used that test standard retrieval as well as multi-hop.

---

> ### Author Rebuttal · Authors · 2025-07-31
>
> Thank you for appreciation of our motivation, method, and discussions. We address the issues you raised below.
>
> ***Issue 1: Missing comparison to strong large-scale embedding models (e.g., baselines in HippoRAG 2).***
>
> Our NeuroPath aims to address the issues of semantic incoherence and noise in graph-based RAG by leveraging a goal-directed, dynamic semantic path tracking mechanism, which is independent of the embedding model size. Experimental results demonstrate that the LLM-driven path tracking mechanism itself brings significant performance gains when compared under the same embedding model conditions (e.g., Contriever and BGE-M3). Moreover, it exhibits scalability across LLMs of different sizes.
>
> We appreciate the reviewer’s suggestion to conduct experiments with larger-scale embedding models. Following the baselines used in HippoRAG2, we selected three 7B-scale embedding models for evaluation, and set the embedding models for both HippoRAG2 and NeuroPath to NV-Embed-v2.
>
> Table 1: Retrieval results using NV-Embed-v2 as the embedding model and GPT-4o-mini as the LLM for both HippoRAG 2 and NeuroPath.
>
> | Method                      | MuSiQue  |          | 2Wiki    |          | HotpotQA |          |
> | --------------------------- | -------- | -------- | -------- | -------- | -------- | -------- |
> | (use GPT-4o-mini)           | R@2      | R@5      | R@2      | R@5      | R@2      | R@5      |
> | gte-Qwen2-7B-instruct       | 48.2     | 63.6     | 66.7     | 74.8     | 75.9     | 89.9     |
> | GritLM-7B                   | 49.6     | 65.8     | 67.2     | 76.3     | 79.1     | 93.3     |
> | NV-Embed-v2                 | 52.7     | 69.3     | 69.1     | 76.5     | **83.7** | 95.3     |
> | HippoRAG 2 (NV-Embed-v2)    | 53.5     | 74.2     | 74.6     | 90.2     | 80.5     | 95.7     |
> | **NeuroPath (NV-Embed-v2)** | **54.1** | **76.0** | **79.3** | **97.3** | 81.3     | **97.0** |
>
> The results show that NeuroPath remains at a leading level and achieves significant gains compared to NV-Embed-v2.
>
>
>
> We also conducted experiments using open-source LLMs Qwen3-8B and Qwen3-4B.
>
> Table 2: Retrieval results after replacing the LLM with Qwen3-8B and Qwen3-4B.
>
> | Method                      | MuSiQue  |          |  2Wiki   |          | HotpotQA |          |
> | :-------------------------- | :------: | :------: | :------: | :------: | :------: | :------: |
> | (use Qwen3-8B)              |   R@2    |   R@5    |   R@2    |   R@5    |   R@2    |   R@5    |
> | HippoRAG 2 (NV-Embed-v2)    |   55.0   |   73.1   |   74.7   |   88.8   |   83.2   |   96.0   |
> | **NeuroPath (NV-Embed-v2)** | **57.7** | **78.2** | **81.5** | **97.5** | **83.6** | **97.0** |
> | (use Qwen3-4B)              |   R@2    |   R@5    |   R@2    |   R@5    |   R@2    |   R@5    |
> | HippoRAG 2 (NV-Embed-v2)    |   55.1   |   73.5   |   74.9   |   88.9   | **83.3** |   96.0   |
> | **NeuroPath (NV-Embed-v2)** | **57.6** | **78.6** | **81.7** | **97.4** |   82.5   | **96.8** |
>
> The results show that NeuroPath still outperform HippoRAG 2 on small-scale open-source models.
>
> We believe smaller embedding models better reflect the performance gap between different RAG methods, particularly graph-based methods, and their sensitivity to embedding quality. In our experiments, other methods showed up to 20% gap on 2WikiMultiHopQA when changing embeddings. In contrast, our method remains stable across all three datasets (Section 4.2, Page 6).
>
> Additionally, Contriever and BGE-M3 require only 1–5 GB of GPU memory to support most tasks, while larger models like NV-Embed-v2 (7B) need up to 26 GB in full precision. In our experiments, even with float16, it still required at least 20 GB, making it impractical for real-world use. We agree that using diverse embedding models helps evaluate robustness, but in future agentic applications, allocating resources to LLMs is likely more beneficial. Our method already achieves substantial gains with lightweight embedding models.
>
>
>
>
>
> ***Issue 2: Does the method generalize beyond multi-hop retrieval?***
>
> We have verified that NeuroPath exhibits strong scalability on both standard retrieval and sense-making tasks:
>
> (i) We have already evaluated the scalability of NeuroPath on standard retrieval tasks (PopQA and NaturalQuestions) in Appendix D (Page 15). The results demonstrate that our NeuroPath still outperforms graph-based methods (achieving 15% and 7% improvements over HippoRAG2 in recall@2 and recall@5, respectively) and remains highly competitive even when compared to iter-based methods like Iter-RetGen (with 14% and 6% gains on the PopQA dataset, respectively).
>
> Table 3: Retrieval performance on standard retrieval tasks. (Page 15)
>
> ||PopQA||NQ||Avg.||
> |-|-|-|-|-|-|-|
> |Method|R@2|R@5| R@2| R@5| R@2| R@5|
> |Contriever| 27.0               | 43.2               | 29.1               | 54.6                | 28.1               | 48.9               |
> | Iter-RetGen| 35.4               | 46.6               | **40.9**           | **68.6**            | 38.1 | **57.6**           |
> | IRCoT| 30.2               | 38.2               | 35.1               | 55.8                | 32.6               | 47.0               |
> | HippoRAG| 36.5\* | **52.7**           | 22.5               | 45.7                | 29.5               | 49.2               |
> | HippoRAG 2| 34.2               | 47.4               | 32.1               | 59.5                | 33.2               | 53.4               |
> | **NeuroPath** | **40.5**           | 49.3\* | 35.8\* | 65.2\* | **38.2**           | 57.2\* |
>
> (ii) In addition to standard and multi-hop tasks, we also validated the scalability of NeuroPath on the sense-making task (the ability to interpret larger, more complex, or uncertain context). Our experiments followed the setup and baselines from the HippoRAG 2, using the NarrativeQA dataset. This dataset requires understanding and reasoning over large-scale narratives, such as long novels.
>
> Table 4: QA performance on sense-making task.
>
> | Method           | NarrativeQA  (F1 Score) |
> | ---------------- | ----------------------- |
> | NV-Embed-v2 (7B) | 24.2                    |
> | GraphRAG         | 20.9                    |
> | LightRAG         | 9.0                     |
> | HippoRAG         | 16.1                    |
> | HippoRAG 2       | 25.2                    |
> | **NeuroPath**    | **32.7**                |
>
> Although not originally designed for sense-making, NeuroPath performs best on NarrativeQA. However, our focus was on factual multi-hop retrieval tasks. Experiments show that NeuroPath can be extended to more complex or uncertain tasks. We attribute this capability to the dynamic tracking process of the LLM, where the model can autonomously choose the most plausible path under uncertain context.

---

> > ### Author Response · Authors · 2025-08-09
> > **Follow-up Before Discussion Period Ends**
> >
> > Dear Reviewer GVGa:
> >
> > I hope this message finds you well. With less than a day remaining in the discussion period, I would like to ensure that the above clarifications and additional experiments have satisfactorily addressed all of your concerns.
> > If you are satisfied, we would be grateful if you could consider updating your score to reflect the discussion outcome.
> >
> > Thank you very much for your time and effort in reviewing our work.
> >
> > Best Regards,
> >
> > The Authors of Submission 11964

---

### Official Review · Reviewer_hmsH · 2025-07-05

**Clarity:** 3
**Significance:** 3
**Originality:** 2
**Rating:** 4
**Confidence:** 3

**Summary:**

The paper introduces NeuroPath, a RAG framework inspired by the path navigation mechanism in neurobiology, specifically hippocampal place cells. The framework aims to improve multi-hop QA by dynamically tracking semantic paths within a KG, enhancing semantic coherence and reducing noise. The method consists of two main components: Dynamic Path Tracking, which selectively filters and expands semantic paths, and Post-retrieval Completion, which refines paths through additional retrieval after the initial path tracking. NeuroPath is shown to outperform existing RAG frameworks, achieving state-of-the-art results on multiple QA datasets with notable improvements in retrieval performance and efficiency.

**Questions:**

1. Could the authors provide further insights into the scalability of NeuroPath for more diverse and large-scale datasets, especially those with noisy or incomplete knowledge graphs?

**Ethical Concerns:**

["NO or VERY MINOR ethics concerns only"]

**Limitations:**

Yes

**Quality:**

3

**Strengths And Weaknesses:**

### **Strengths**:

1. The use of neurobiological inspiration (specifically hippocampal place cells) in designing the path-tracking mechanism is a creative and original approach. This neurobiology-inspired method introduces a dynamic, goal-directed way to improve multi-hop reasoning, addressing the issues of noise and semantic incoherence prevalent in traditional RAG methods.

2. The experiments demonstrate that NeuroPath outperforms SOTA models on several multi-hop QA datasets (MuSiQue, 2Wiki MultiHopQA, HotpotQA), achieving substantial improvements in recall and retrieval accuracy, while also reducing token consumption by 22.8%.

3. The paper provides a detailed description of the methodology, including the rationale behind the neurobiological inspiration, the steps of path tracking, and post-retrieval completion, making the framework easy to understand.

### **Weaknesses:**

1. While the framework performs well across different LLMs and models, its scalability and generalization to much larger, more diverse datasets (e.g., noisy, incomplete, or domain-specific data) could be explored further.

2. The reliance on coreference resolution for entity linking and path expansion could lead to errors when dealing with less structured or unclean data, which the authors do not fully address in the paper.

3. While the method reduces token consumption compared to iter-based methods, it may still be resource-intensive due to the dynamic nature of the path-tracking and post-retrieval steps, potentially limiting its application in large-scale, real-time systems.

4. My main concern of the usefulness of this method is the reliance on high-quality KGs, which limits the value of the method in real-world scenarios.

---

> ### Author Rebuttal · Authors · 2025-07-31
>
> Thank you for appreciating the clear motivation, detailed description, and substantial improvements of NeuroPath. Below, we will address the weaknesses you point out.
>
> ***Weakness 1: Should explore the framework's scalability and generalization***
>
> 1. We have already evaluated the scalability of NeuroPath on standard retrieval tasks (PopQA and NaturalQuestions) in Appendix D (Page 15). The results demonstrate that our NeuroPath still outperforms graph-based methods and remains highly competitive compared to iter-based methods.
>
>    Table 1: Retrieval performance on standard retrieval tasks. (Page 15)
>
>    || PopQA||NQ|| Avg.||
>    | - | - | - | - | - | - | - |
>    | Method | R@2 | R@5  | R@2 | R@5| R@2 | R@5 |
>    | Contriever| 27.0 | 43.2 | 29.1 | 54.6 | 28.1| 48.9  |
>    | Iter-RetGen   | 35.4| 46.6| **40.9** | **68.6**  | 38.1 | **57.6** |
>    | IRCoT | 30.2 | 38.2 | 35.1 | 55.8| 32.6| 47.0|
>    | HippoRAG | 36.5\* | **52.7** | 22.5 | 45.7 | 29.5 | 49.2 |
>    | HippoRAG 2 | 34.2 | 47.4| 32.1| 59.5  | 33.2 | 53.4|
>    | **NeuroPath** | **40.5**  | 49.3\* | 35.8\* | 65.2\* | **38.2** | 57.2\* |
>
>
>
> 2. We conducted additional experiments on the MultiHop-RAG benchmark, a challenging dataset specifically designed for evaluating multi-hop retrieval and reasoning. MultiHop-RAG contains 2,556 questions requiring 2 to 4 hops of reasoning. It features longer candidate documents and noisier supporting evidence.
>
>    Table 2: Comparison of the average number of tokens per document.
>
>    || MuSiQue | 2Wiki | HotpotQA | MultiHop-RAG|
>    | - | - | - | - | - |
>    | Average tokens per document |110|105|128| 2,289 (chunk to 512) |
>
>    We selected 1,000 questions for retrieval, and the results are as follows:
>
>    Table 3: Retrieval results on MultiHop-RAG.
>
>    || MultiHop-RAG (Contriever) | MultiHop-RAG (BGE) |
>    | - | - | - |
>    | Method| R@2 / R@5| R@2 / R@5|
>    | Contriever| 9.5 / 20.4 | - |
>    | BGE-M3| - | 25.4 / 44.7 |
>    | Iter-RetGen| 17.8 / 30.9| **28.8** / 45.8 |
>    | HippoRAG 2| 15.7 / 29.5| 21.4 / 40.8 |
>    | **NeuroPath** | **23.7** / **39.0** | 26.1 / **46.8**|
>
>    Our method outperforms typical baselines, especially when using the smaller Contriever. Notably, other methods still exhibit sensitivity to the choice of embedding model, whereas NeuroPath shows relatively stable performance differences between Contriever and BGE-M3 (consistent with the conclusions in Section 4.2, Page 6).
>
>
>
> 3. In addition to multi-hop and standard tasks, we also evaluated the scalability for the sense-making task (the ability to interpret larger, more complex, or uncertain context). We followed the setup and baselines from the HippoRAG 2, using the NarrativeQA dataset. This dataset requires understanding and reasoning over large-scale narratives, such as long novels.
>
>    Table 4: QA performance on NarrativeQA.
>
>    | Method| NarrativeQA (F1 Score) |
>    | - | - |
>    | NV-Embed-v2 (7B) | 24.2|
>    | GraphRAG | 20.9|
>    | LightRAG | 9.0 |
>    | HippoRAG | 16.1|
>    | HippoRAG 2| 25.2 |
>    | **NeuroPath** | **32.7**|
>
> The results show that NeuroPath can be extended to more complex or uncertain tasks. We attribute this capability to the dynamic path tracking process of the LLM, where the model can autonomously choose the most likely path under uncertain context.
>
>
>
> ***Weakness 2: Reliance on coreference resolution may cause errors on less structured or unclean data.***
>
> While fundamentally improving graph construction is not the goal of our work, we mitigate coreference-related errors at runtime.
>
> The quality of generated KG data depends on the LLM's performance. While existing methods depend on multi-step LLM extraction, we simplify this with one-pass KG extraction and efficient embedding-based coreference resolution.
>
> Although we simplified the process to achieve faster indexing, we still adopt a two-stage solution to address errors in coreference linkage:
>
> 1. Path tracking: For nodes with the same surface form but different referents, each path expansion step collects a series of triple fragments that carry relational context. The LLM uses these fragments to disambiguate and filter irrelevant coreferences.
> 2. Post-retrieval completion: If seed nodes or their coreference sets fail to capture key nodes during expansion, the final path may lack information. We address this with an additional embedding-based retrieval step using the output of the last expansion, which includes requests for missing information. (Ablation results: Section 4.4, Page 8.)
>
> Despite the simplified KG construction, our NeuroPath still outperforms other graph-based baselines, demonstrating the strength of our retrieval framework.
>
> We also verify whether LLMs can benefit from identifying phrases in the coreference set by changing the number k of expanded coreferent phrases (0, 1, 5).
>
> Table 5: Impact of the number k of expanded coreferent phrases on retrieval.
>
> || MuSiQue|| 2Wiki || HotpotQA ||
> |-|-|-|-|-|-|-|
> |      | R@2     | R@5  | R@2   | R@5  | R@2      | R@5  |
> | k=5  | 48.0    | 62.7 | 77.2  | 92.5 | 75.6     | 90.4 |
> | k=1  | 47.8    | 63.7 | 76.2  | 91.3 | 74.0     | 89.2 |
> | k=0  | 47.2    | 63.1 | 75.5  | 90.6 | 73.3     | 88.7 |
>
> In our entity extraction prompts (Figure 12, Page 22), we explicitly instruct the LLM to use noun phrases with explicitly referents. As a result, it tends to extract consistent phrases for the same entity, which explains why the performance gains from k = 0, 1, and 5 are relatively small. Nonetheless, the results show that using coreference sets still brings improvements, indicating that the LLM can recognize and leverage them.
>
>
>
> ***Weakness 3: Method may still be resource-intensive due to the dynamic nature of the path-tracking and post-retrieval steps.***
>
> This insightful comment aligns with our goal of balancing performance and cost. We respectfully clarify that the main source of resource consumption (i.e., LLM calls) lies in the path tracking stage. The post-retrieval completion stage simply extracts the output from the final step of path tracking and performs fast vector retrieval. Its purpose is to compensate for any missing information that may have been overlooked during the path tracking process. In fact, it was our intentional resource control that resulted in a 22.8% reduction in token consumption over the iter-based methods.
>
> In Dynamic Path Tracking stage, we use 2 strategies to control the token cost:
>
> 1. Candidate paths are pruned before being fed into the LLM. We use an embedding model to compute the similarity between each candidate path and the query, retaining only the top 30 paths. (default setting)
> 2. The LLM is used to further select valid paths, in order to prevent the expansion of irrelevant paths in the next step (if any).
>
> Table 6: Ablation study of the pruning strategy. (Section 4.4, Page 8)
>
> || MuSiQue ||| 2Wiki ||| HotpotQA |||
> |-|-|-|-|-|-|-|-|-|-|
> || R@2| R@5| Token (k)|R@2|R@5|Token (k)| R@2|R@5|Token (k)|
> | w/o pruning    | 48.7    | 63.8 | 2891| 76.8  | 92.0 | 1883              | 75.7     | 90.8 | 2504|
> | default (p=30) | 48.0 | 62.7 | $\downarrow$45.7% | 77.2  | 92.5 | $\downarrow$8.4%  | 75.9 | 90.1 | $\downarrow$39.8% |
> | p=20           | 47.3 | 62.2 | $\downarrow$52.6% | 76.5  | 90.8 | $\downarrow$17.3% | 74.9 | 89.4 | $\downarrow$47.0% |
>
> Our ablation studies (Section 4.4, Page 8) systematically evaluate the impact of path pruning on computational efficiency. Specifically, we compare three configurations: 1) no pruning, 2) retaining top-20 paths, and 3) retaining top-30 paths (default). The results demonstrate that:
>
> 1. Pruning to 20 paths reduces token consumption by 17.3%-52.6%
> 2. Our default setting (30 paths) achieves a 8.4%-45.7% reduction in token usage
>     Notably, both pruning configurations maintain comparable recall performance, with the 30-path setting even showing slight improvements in some cases.
>
> In real-world applications, the maximum number of expansion hops and pruning thresholds can be flexibly adjusted.
>
> As noted in Appendix H (Page 19), LLM calls are the runtime bottleneck. Nevertheless, our method demonstrates the feasibility of leveraging LLMs for path tracking to address semantic incoherence and noise issues, leading to substantial improvements in retrieval quality for multi-hop queries. It also shows scalability on different tasks, such as standard factual retrieval task and sense-making task. Based on the above, we consider the tradeoff between performance and cost to be reasonable.
>
>
>
> ***Weakness 4: The main concern is its reliance on high-quality KGs, which may limit real-world applicability.***
>
> We appreciate the reviewer’s concern about how KG quality may affect NeuroPath in real-world scenarios. To enhance real-world applicability, we will extend NeuroPath in two directions:
>
> (a) Task-adaptive structured information processing. While the current KG directly reflects raw document content, higher-level tasks (e.g., abstraction or summarization) may require restructuring knowledge granularity. This flexibility will broaden task scalability.
>
> (b) Efficient high-quality path expansion. Building upon optimized KGs, we will explore accelerated path-tracking mechanisms to alleviate LLM bottlenecks while maintaining retrieval precision.
>
>
>
> ***Question: Explore the framework's scalability and generalization for more diverse and large-scale datasets.***
>
> As we detailed in our response to Weakness 1, our framework not only extends to noisier and more challenging multi-hop retrieval tasks, but also generalizes to standard retrieval tasks and more complex, uncertain sense-making tasks. Experimental results demonstrate its strong scalability and performance. We attribute this to the autonomy of the LLM in dynamic tracking.

---

> > ### Comment · Reviewer_hmsH · 2025-08-04
> >
> > Thank you for your detailed reply. Please consider strengthening the paper’s organization, and experimental comparisons in the future version. I will keep my current score in this round.

---

### Official Review · Reviewer_y7M1 · 2025-07-07

**Clarity:** 4
**Significance:** 4
**Originality:** 4
**Rating:** 6
**Confidence:** 4

**Summary:**

The paper proposes NeuroPath, a biological inspired RAG for knowledge reasoning over multi-hop (multi-document) questions. The paper is a good followup of how thinking is organized into semantic units that are then connected based on semantic traversal towards a goal. A query drives the logical connections the brain follows to derive the answer. Based on this, the paper extends earlier works short-comings 1)  topological (structural) aggregation of subgraph in PathRAG/LightRAG 2) Page rank based subgraph selections in HippoRAG, which are noisy and not as semantically aligned to the goal target. The proposed NeuroPath achieves this with expanding the search from seed nodes to the target through semantic reasoning paths using prompt-based LLMs and retrieval in this iterative (multi-step) RAG framework.

They also compare against topological baselines (HippoRAG, PathRAG, LightRAG) and iterative based retrieval baselines (Iter-RAG, and IterCoT) covering the full space of methods relevant to the comparison here. Their work provides two strong retrievers (Contriever and BGE-m3) along with 4 LLM models of varying sizes, making the study comprehensive.

**Questions:**

None

**Ethical Concerns:**

["NO or VERY MINOR ethics concerns only"]

**Final Justification:**

Ack the rebuttal. Same review/score as before.

**Limitations:**

yes

**Quality:**

4

**Strengths And Weaknesses:**

Strengths:
1. **Motivation and contextualization with the latest research in RAG**: The neurologically inspired framework is inspiring, and it is contextualized very effectively with the recent advances on RAG for multi-document reasoning. It provides a valuable solution to the recent research short-comings 1)  topological (structural) aggregation of subgraph in PathRAG/LightRAG 2) Page rank based subgraph selections in HippoRAG, which are noisy and not as semantically aligned to the goal target. The proposed NeuroPath achieves this with expanding the search from seed nodes to the target through semantic reasoning paths using prompt-based LLMs and retrieval in this iterative (multi-step) RAG framework, with properly motivated and analyzed cases (e.g. appx F.3).
2.  **Experimental evaluations and analysis**: The evaluations are comprehensive and robust. They compare against topological baselines (HippoRAG, PathRAG, LightRAG) and iterative based retrieval baselines (Iter-RAG, and IterCoT) covering the full space of methods relevant to the comparison here. Their work provides two strong retrievers (Contriever and BGE-m3) along with 4 LLM models of varying sizes, making the study comprehensive. The evaluation and analysis of the failures are insightful and make it clear how different methods behave rather than relying solely on numerical results (including retrieval recall, and EM/F1).
3. **Reproducibility** - Clear evaluations, details of the algorithm in the appendix and prompts, make it easily reproducible.

Weakness:
1. **Graph construction offline scale**: IterRet-Gen (Iterative RAG) is strong as a baseline method and with a good retriever and generator, it can expand the context and get the multi-documents needed to derive the answer iteratively. It has lower complexity than the proposed method - which requires Graph Construction (with LLM) to pre-structure all the documents (at a large scale this is costly). The proposed method results outperform IterRet-Gen with some margin, but the graph construction complexity at scale might mitigate the practical benefit over Iter-RAG methods. Yet, the neurologically inspired formulation and contextualization with recent research in RAG, and the detailed evaluations and insights make the proposed paper worthwhile for advancing the field or summarizing these solutions to multi-document RAG reasoning.

---

> ### Author Rebuttal · Authors · 2025-07-31
>
> Thank you for recognizing the novelty and experimental rigor of NeuroPath. We provide the following responses and perspectives regarding the weakness you pointed out.
>
> ***Weakness: Graph construction offline is costly.***
>
> We agree that offline KG construction incurs upfront cost. However, NeuroPath and other RAG systems are designed for scenarios requiring repeated queries over static corpora. Here, one-time indexing amortizes cost, while online retrieval achieves 22.8% lower token consumption vs. iter-based methods (Appendix G, Page18).
>
> Table 1: Comparison of graph indexing time with graph-based methods. (Page 19)
>
> |                | HippoRAG | HippoRAG 2 | LightRAG | PathRAG | NeuroPath |
> | -------------- | -------- | ---------- | -------- | ------- | --------- |
> | Time (minutes) | 64       | 62         | 52       | 72      | 39        |
>
> Our KG construction is 1.5× faster than graph/path-based RAG baselines (39 min vs. 52–72 min for HippoRAG/LightRAG/PathRAG on 2WikiMultihopQA, Table 11, Page 19) through:
>
> 1. Single-pass KG extraction: LLM generates entities and relations per document in one call (vs. multi-stage pipelines in baselines).
> 2. Embedding-based coreference resolution and similarity computation: Avoids LLM calls.
>
> Although the graph construction stage inevitably incurs some cost, experiments show that NeuroPath brings significant gains over graph-based and iter-based methods under different LLMs and embedding models, demonstrating the effectiveness of our framework.
>
> It is worth noting that finer-grained path-based retrieval is less sensitive to the performance of the embedding model (Section 4.2). Even small embedding models (e.g., Contriever) can effectively capture path semantics. In contrast, document-level retrieval in iter-based methods is highly sensitive to embedding model performance, with up to a 20% performance gap observed across different embedding models on the 2WikiMultiHopQA dataset (Table 1, Page 7).
>
> Additionally, we further validated that NeuroPath can be extended beyond multi-hop retrieval to other tasks:
>
> 1. Standard retrieval tasks (PopQA and NaturalQuestions)
>
>    We validated the generalizability of NeuroPath on standard retrieval tasks in Appendix D (Page 15). The results demonstrate that our NeuroPath still outperforms graph-based methods (achieving 15% and 7% improvements over HippoRAG2 in recall@2 and recall@5, respectively) and remains highly competitive even when compared to iter-based methods like Iter-RetGen (with 14% and 6% gains on the PopQA dataset, respectively).
>
>    Table 2: Retrieval performance on standard retrieval tasks. (Page 15)
>
>    |               | PopQA    |          | NQ       |          | Avg.     |          |
>    | ------------- | -------- | -------- | -------- | -------- | -------- | -------- |
>    | Method        | R@2      | R@5      | R@2      | R@5      | R@2      | R@5      |
>    | Contriever    | 27.0     | 43.2     | 29.1     | 54.6     | 28.1     | 48.9     |
>    | Iter-RetGen   | 35.4     | 46.6     | **40.9** | **68.6** | 38.1\*   | **57.6** |
>    | IRCoT         | 30.2     | 38.2     | 35.1     | 55.8     | 32.6     | 47.0     |
>    | HippoRAG      | 36.5\*   | **52.7** | 22.5     | 45.7     | 29.5     | 49.2     |
>    | HippoRAG 2    | 34.2     | 47.4     | 32.1     | 59.5     | 33.2     | 53.4     |
>    | **NeuroPath** | **40.5** | 49.3\*   | 35.8\*   | 65.2\*   | **38.2** | 57.2\*   |
>
> 2. More complex, uncertain retrieval scenarios such as sense-making tasks (NarrativeQA).
>
>    We follow the experimental setup and baselines from the HippoRAG 2 paper, using NV-Embed-v2 as the embedding model. The results show that our method surprisingly outperforms the other baselines. Although our method was not originally designed for sense-making tasks, we seem to be able to demonstrate scalability. We attribute this to LLM actively choosing possible paths during path tracking when facing ambiguous or uncertain context.
>
>    Table 3: QA performance on NarrativeQA (sense-making tasks)
>
>    | Method           | NarrativeQA (F1 Score) |
>    | ---------------- | ---------------------- |
>    | NV-Embed-v2 (7B) | 24.2                   |
>    | GraphRAG         | 20.9                   |
>    | LightRAG         | 9.0                    |
>    | HippoRAG         | 16.1                   |
>    | HippoRAG 2       | 25.2                   |
>    | **NeuroPath**    | **32.7**               |
>
> We appreciate your recognition of these strengths and hope this clarifies the balance between efficiency, scalability, and performance.

---

### Note · Authors · 2025-08-12

We sincerely thank all reviewers and AC for their time and constructive feedback. Below, we summarize the consistent consensus on our work and our clarifications regarding the key concerns.

1. Consensus on contributions and strengths

* **Novelty & significance.** A neurobiology-inspired, goal-directed semantic path tracking framework addressing semantic incoherence and noise in multi-hop retrieval. The framework is well-motivated and timely to solve the problems faced by current graph-based RAG systems.
* **Strong empirical results.** Consistent gains over baselines via experiments, ablations and case studies.
* **Understandable & reproducible.** Detailed methodology, workflow pseudocode, prompts, and released code.

2. Common and specific key concerns

* **Generality & scalability.** In response to Reviewers hmsH and GVGa, additional results on PopQA and NQ (Appendix D), NarrativeQA, and noisy MultiHop-RAG show robustness across standard, sense-making, and multi-hop tasks.
* **Cost & efficiency.** In response to Reviewers y7M1, hmsH, and QcTf, we provide detailed quantification and comparison in Appendix G. KG construction is 1.5× faster and reduces tokens by 31.1%. Path-level retrieval and pruning reduce tokens by 22.8%. In addition, ablation studies evaluate the effectiveness of the pruning strategy in reducing token consumption while maintaining performance.
* **Large-scale embedding baseline.** In response to Reviewer GVGa, we added experiments with large-scale embedding model NV-Embed-v2 to confirm that NeuroPath remains SOTA, including on smaller LLMs.
* **HotpotQA's shortcut analysis & complex benchmark.** In response to Reviewer QcTf, we analyzed HotpotQA's shortcut issue and compared our method with iter-based methods. Our method is better suited for complex multi-hop QA and shows scalability across tasks. We added experiments on the more challenging MultiHop-RAG dataset, where NeuroPath remains superior.

* **Clarity of motivation and methodology.** In response to Reviewer QcTf, we provided concrete definitions, intuitive examples, and reproducible details in both the paper and the rebuttal (acknowledged as clear by other reviewers).

In summary, we have carefully responded to each concern and believe that most have been addressed. NeuroPath offers a novel and practical contribution to multi-hop retrieval, achieving SOTA performance and broad scalability. We respectfully request the positive consideration of both the reviewers and the AC.

---

### Decision · Program_Chairs · 2025-09-17

**Decision:**

Accept (poster)

**Comment:**

The paper introduces NeuroPath, a biologically inspired retrieval-augmented generation (RAG) framework for multi-hop (multi-document) question answering. The approach is motivated by how human reasoning organizes knowledge into semantic units and connects them through goal-directed semantic traversal. A query guides the construction of logical connections, mimicking how the brain derives answers.

Building on this intuition, NeuroPath addresses the limitations of prior work: (1) the purely topological subgraph aggregation used in PathRAG and LightRAG, and (2) the PageRank-based subgraph selection in HippoRAG, which is often noisy and less semantically aligned with the query. Instead, NeuroPath expands the search from seed nodes toward the target via semantic reasoning paths, implemented through prompt-driven LLMs and iterative retrieval in a multi-step RAG framework.

The evaluation compares NeuroPath against both topological baselines (HippoRAG, PathRAG, LightRAG) and iterative retrieval baselines (Iter-RAG, IterCoT), covering the full spectrum of relevant methods. The study is made comprehensive by combining two strong retrievers (Contriever and BGE-m3) with four LLMs of different scales.

In the rebuttal, the authors have provided comprehensive experiments to alleviate the reviewers' concerns. Three reviewers present positive ratings on the paper; the other reviewer gave a score of 3. The negative reviewer did not provide further feedback after the authors' responses, and I believe the weak points can be easily solved in the final version. I tend to recommend acceptance.